# Widespread potential for phototrophy and convergent reduction of lifecycle complexity in the dimorphic order *Caulobacterales*

Joel Hallgren [1], Jennah E. Dharamshi [2,5], Alejandro Rodríguez-Gijón [2], Julia Nuy [2,6], Sarahi L. Garcia [2,3,4] & Kristina Jonas [1] ✉

Model bacteria are fundamental for research, but knowledge about their ecology and evolution is often limited. Here, we establish an evolutionary and ecological context for the model organism *Caulobacter crescentus*—an alpha-proteobacterium intensively studied for its dimorphic lifecycle. By analyzing the phylogenetic relatedness and genetic potential of hundreds of *Caulobacterales* species, we reveal substantial diversity regarding their environmental distribution, morphology, cell development, and metabolism. Our work provides insights into the evolutionary history of morphological features such as the cell curvature determinant crescentin and uncovers a striking case of convergent loss of traits for cellular dimorphism among close relatives of *C. crescentus*. Moreover, we find that genes for phototrophy are widespread across *Caulobacterales* and that the new genus *Acaudatibacter*, described here, includes the first reported *Caulobacterales* lineage with photoautotrophic potential. Our study advances our understanding of an environmentally widespread bacterial order and sheds light on the evolution of fundamental prokaryotic features.

Most bacteriological research relies on a limited number of model bacteria, which often represent large taxonomic groups[1,2]. The bias towards a few species, primarily human pathogens, leaves the vast majority of bacteria understudied, greatly narrowing the scope of known prokaryotic biology. A current challenge in microbiology is to integrate existing knowledge about model bacteria with the immense and unexplored bacterial diversity revealed by the recent advancement of cultivation-independent techniques, such as metagenomics[3].

One example of a well-studied model bacterium is *Caulobacter crescentus* (also known as *C. vibrioides*[4]). Since its isolation in the 1960s, it has been studied intensively for its complex lifecycle and today serves as one of the forefront models for bacterial cell and developmental biology[5,6]. The *C. crescentus* lifecycle is dimorphic, comprising two morphologically distinct stages: a brief motile dispersive 'swarmer' stage and a mature sessile reproducing 'stalked' stage[7]. The stalked cell is named after its thin cell envelope extension (prostheca), which it uses to attach itself to a surface with its holdfast adhesin. Stalked cells always divide asymmetrically to generate swarmer cells from their stalk-distal cell pole. Newborn swarmer cells are capable of exploring their environment using their flagellum, chemotaxis machinery, and adhesive type IV pili (T4P), but cannot reproduce until they have irreversibly differentiated into sessile stalked cells[7]. This cell differentiation event, which is an integral part of the *Caulobacter* cell cycle, along with the sophisticated spatiotemporal

[1]Department of Molecular Biosciences, The Wenner-Gren Institute, Science for Life Laboratory, Stockholm University, Stockholm, Sweden. [2]Department of Ecology, Environment and Plant Sciences, Science for Life Laboratory, Stockholm University, Stockholm, Sweden. [3]Institute for Chemistry and Biology of the Marine Environment (ICBM), Carl von Ossietzky Universität Oldenburg, Oldenburg, Germany. [4]Helmholtz Institute for Functional Marine Biodiversity at the University of Oldenburg (HIFMB), Oldenburg, Germany. [5]Present address: Department of Organismal Biology, Program in Systematic Biology, Uppsala University, Uppsala, Sweden. [6]Present address: Environmental Metagenomics, Research Center One Health, University of Duisburg-Essen, Essen, Germany. ✉e-mail: kristina.jonas@su.se

mechanisms underlying its asymmetric cell division, has attracted extensive research during the past 50 years, which has fundamentally shaped our views on bacterial cell organization and development[5,6].

Despite decades of molecular research, the precise ecological relevance and evolutionary history of *C. crescentus* and its dimorphic lifecycle remain unknown. Research on *C. crescentus* has relied on two lab-adapted strains (CB15/NA1000) derived from the same pond water isolate[8,9]. Accordingly, *Caulobacter* was long assumed to be primarily found in oligotrophic aquatic environments. However, recent reports have shown that the genus *Caulobacter* is also prevalent in terrestrial environments, including soil and in association with plant material[10]. *Caulobacter* belongs to the alphaproteobacterial family *Caulobacteraceae*, a lineage of aerobic heterotrophs that includes biodegraders, plant commensals, and opportunistic human pathogens[11–15]. However, the diversity within the *Caulobacteraceae* and other families of the order *Caulobacterales* remains poorly studied, despite their ubiquity in nature. Thus, it is unclear to what extent the characteristic features of *C. crescentus* are shared with its relatives.

Recent advances in cultivation-independent methods, such as genome-resolved metagenomics, have enabled the reconstruction of genomes from a greatly expanded diversity of environmental bacteria, including many uncharacterized *Caulobacterales* clades. Here, we use publicly available metagenome-assembled genomes (MAGs) and isolate genomes, to investigate the diversity and evolutionary history of the *Caulobacterales* order, focusing on *Caulobacteraceae*, its largest family. Our work reveals several unexpected features of this ubiquitous group of bacteria. Notably, our results indicate the convergent loss of flagellar motility and dimorphism among independent *Caulobacteraceae* lineages, helping to define core modules for cellular dimorphism. Moreover, we find that phototrophy is widespread across the order *Caulobacterales* (-10% species), and we identify the first reported *Caulobacterales* lineage with photoautotrophic potential in the new genus *Acaudatibacter*, revealing a previously unrecognized potential role of *Caulobacterales* members as primary producers. Together, our work illuminates the diversity and evolution of an environmentally widespread bacterial order and provides new perspectives on the evolution of bacterial lifecycle complexity.

## Results

### An updated species phylogeny of the *Caulobacterales*

To explore the genomic and environmental diversity within the family *Caulobacteraceae* and its position within the order *Caulobacterales*, we reconstructed an updated species phylogeny using all available high-quality *Caulobacterales* genomes (see "Methods"). To achieve this, we grouped genomes into species clusters (≥95% average nucleotide identity; ANI)[16,17] and kept one representative genome per species (Supplementary Data 1a), resulting in 347 *Caulobacterales* species, of which 107 are currently described. We then used the representative genomes to infer a maximum-likelihood (ML) phylogeny using 72 previously curated genes conserved across *Alphaproteobacteria*[18], resulting in a robust species phylogeny with high non-parametric bootstrap support (npBS) for most clades (Fig. 1a and Supplementary Fig. S1). In further support of the inferred phylogenetic relationships, the tree was highly concordant with an alternative, less refined ML phylogeny of 117 alphaproteobacterial genes and with an ML phylogeny inferred for concatenated 16S and 23S rRNA genes (Supplementary Figs. S2 and S3).

In the species tree, the first-described *Caulobacteraceae* genera *Asticcacaulis*, *Brevundimonas*, *Caulobacter*, and *Phenylobacterium*, formed a distinct clade alongside several uncharacterized genus-level groups (Fig. 1a and Supplementary Fig. S1). Notably, the clade did not include the genera *Aquidulcibacter*, *Pseudaquidulcibacter*, and *Terricaulis*, which were recently assigned to *Caulobacteraceae*[19–21]. Instead, these genera formed a clade (100% npBS) that clustered sister to the family *Hyphomonadaceae* (98% npBS), together with *Vitreimonas flagellata*[22] (*Hyphomonadaceae*), indicating that they do not belong to the family *Caulobacteraceae*. We instead propose the name *Aquidulcibacteraceae* fam. nov. for this clade (description found in Supplementary Note 1). Based on our robust and comprehensive species phylogenies (Supplementary Figs. S1 and S2), we propose additional revisions to *Caulobacterales* taxonomy, including the transfer of the recently described species *Phenylobacterium montanum*[23], which consistently clustered outside of *Phenylobacterium* and *Caulobacter*, into the new *Caulobacteraceae* genus *Poindextera* gen. nov. (*Poindextera montana* comb. nov.) (Supplementary Notes 1 and 2). Henceforth, we will refer to the clade composed of *Asticcacaulis*, *Brevundimonas*, *Caulobacter*, *Phenylobacterium*, and *Poindextera* gen. nov. as *Caulobacteraceae*, excluding members of *Aquidulcibacteraceae* fam. nov.

The refined *Caulobacteraceae* family comprised 206 species, of which 152 are undescribed, and was notably more species-rich than the other four *Caulobacterales* families (140 species in total) (Fig. 1a). It was split into two major clades: one including the genera *Asticcacaulis*, *Brevundimonas*, and the genus-level clade "17J80-11" (100% npBS), and the other including *Caulobacter*, *Phenylobacterium*, *Poindextera* gen. nov., as well as the genus-level clade "Palsa-881" and six other uncharacterized genera (99% npBS) (Fig. 1a and Supplementary Fig. S1). Due to its distinct features described below, we renamed "Palsa-881" to *Acaudatibacter* gen. nov. according to the SeqCode[24] nomenclature for uncultivated microorganisms (available under the accession ID seqco.de/r:9aocwnme; Supplementary Notes 3 and 4).

### Environmental distribution of *Caulobacterales*

To investigate *Caulobacterales* environmental distribution, we manually compiled and curated metadata (see "Methods") for all 347 species-cluster genome representatives (Fig. 1b and Supplementary Data 2). We found that *Caulobacteraceae* differs from other *Caulobacterales* families in several regards. First, while the other families were predominantly aquatic, *Caulobacteraceae* included a high proportion (32.4%) of terrestrial lineages (Fig. 1b and Supplementary Fig. S5a). Second, only 5.8% of *Caulobacteraceae* species representatives came from marine/saline environments, while the other *Caulobacterales* families (except *Aquidulcibacteraceae* fam. nov.) were almost exclusively associated with such habitats (Fig. 1b and Supplementary Fig. S5b). Third, unlike other families of the order *Caulobacterales*, the *Caulobacteraceae* family contained a considerable fraction of species associated with plant and animal hosts. Using the IMNGS 16S rRNA gene amplicon database[25], we similarly found that *Caulobacteraceae* most frequently had high relative abundance (≥1%) in environments associated with plants and their roots, as well as freshwater, soil, and wastewater (Supplementary Fig. S5c and Supplementary Data 3). *Caulobacteraceae* sequences were also prevalent in studies of marine, gut, and fecal environments, but less frequently at high relative abundance.

Within the genus *Caulobacter*, nearly all species came from aquatic freshwater environments or from terrestrial environments closely associated with plants (Fig. 1b, Supplementary Fig. S6a, and Supplementary Data 2). Terrestrial plant-associated *Caulobacter* species had considerably larger genomes than aquatic species (Fig. 1c and Supplementary Fig. S6b). This is consistent with a previous report for *Caulobacter*[10] and observations that plant-inhabiting *Pseudomonadota* (*Proteobacteria*) in general have larger genomes[26–28], and suggests that distinct *Caulobacter* lineages are specialized to either plant-associated or aquatic lifestyles. Indeed, terrestrial *Caulobacter* genomes encoded several virulence and host colonization factors rare among aquatic species, including type I (*hlyBD*), type III, and type VI secretion systems, and the sulfonate transport system *ssuABC* (Supplementary Fig. S6a), all suggested mediators of plant–bacterial interactions[29–35].

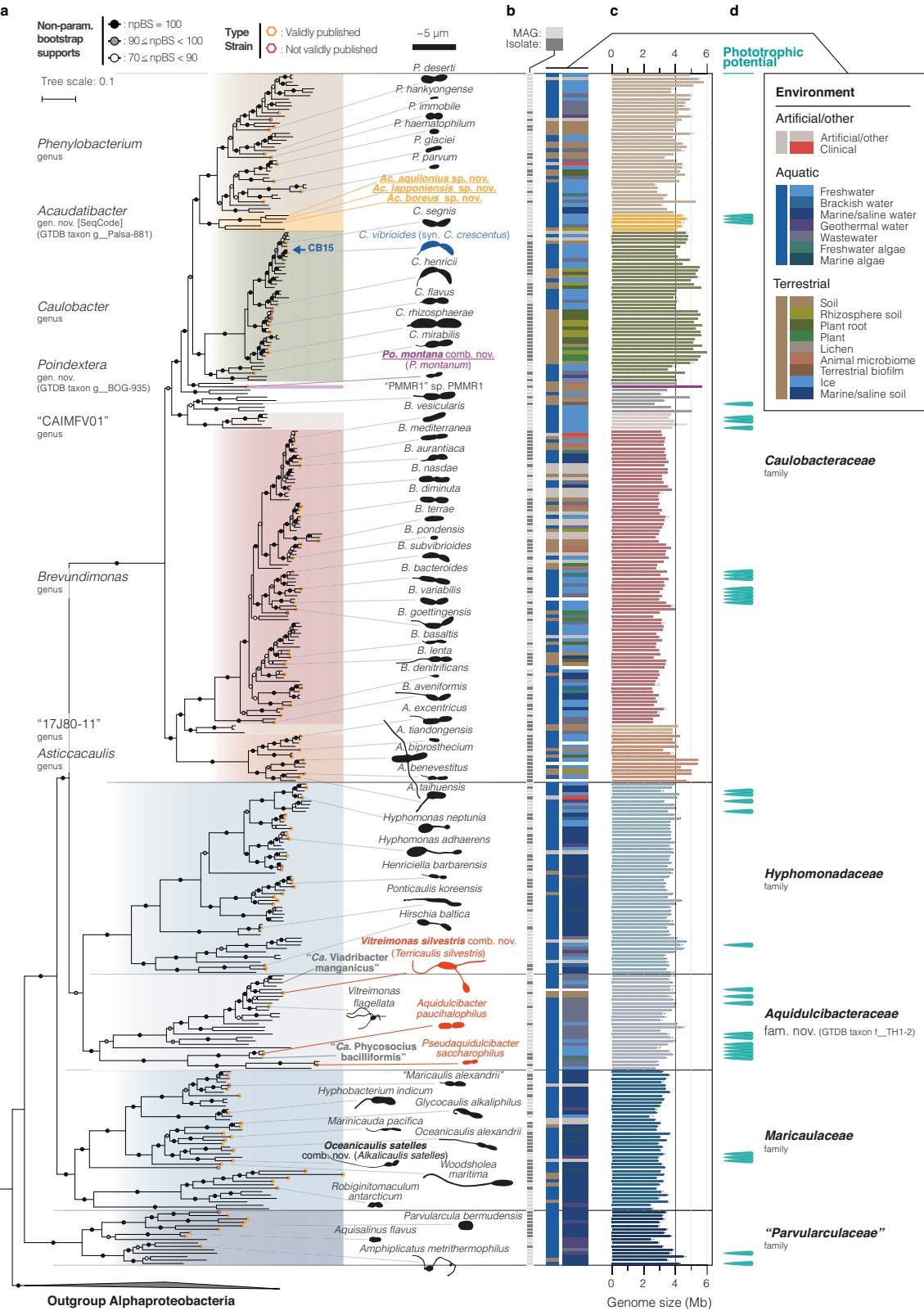

## Convergent losses of dimorphism-related genes in *Caulobacteraceae*

Having established a robust species phylogeny and the environmental distribution of *Caulobacteraceae*, we used these insights to assess the extent to which key features of the model organism *C. crescentus*, such as its well-characterized lifecycle, are representative of the family. During cell development, an obligate motile-to-sessile transition precedes reproduction, and during each cell division, the sessile stalked cell generates motile flagellated offspring (Fig. 2a). In line with this canonical dimorphic lifecycle, most *Caulobacteraceae* genomes had large sets of genes known to mediate flagellar motility and chemotaxis (Fig. 2b and Supplementary Data 5b, c). However, strikingly, four genomes completely lacked such genes: (i) the immotile soil isolate *Phenylobacterium immobile* E[T 36] and (ii) three freshwater MAGs

**Fig. 1 | Species phylogeny and environmental distribution of the order *Caulobacterales*. a** ML concatenated phylogenetic tree of 72 conserved single-copy alphaproteobacterial marker genes[18] inferred using IQ-TREE[109] with the LG + C60 + F + R model of evolution and 100 non-parametric bootstraps (alignment length of 25448 amino acids). The tree was inferred for genome representatives of 347 species clusters (ANI ≥ 95%) from the order *Caulobacterales* (following GTDB taxonomy), the model strain *Caulobacter crescentus* CB15, and five outgroup non-*Caulobacterales Alphaproteobacteria* for rooting. *Poindextera* gen. nov., *Caulobacteraceae* genera with ≥ 3 species clusters, and other *Caulobacterales* families are indicated with colored boxes in the phylogeny. npBS ≥ 70 are indicated with shaded circles, as outlined in the top-left corner, and the thin scale bar indicates the number of substitutions per site in the phylogeny. Genomes of validly described species type strains are marked with orange hexagons (LPSN[125], accessed March 21, 2025). Schematic drawings of cell silhouettes of selected species highlight morphological diversity and are drawn roughly to scale (thick scale bar) from microscopy images (Supplementary Fig. S4) and published material (references are listed in Supplementary Table S1). Red cell outlines highlight species misclassified as *Caulobacteraceae*. The tree is presented alongside all species names and genome accessions in Supplementary Fig. S1. **b** Color-coded genome metadata. Column 1 distinguishes MAGs in light gray from isolate genomes in dark gray. Column 2 indicates the overarching environmental source for each genome, and column 3 provides more details on the sampling/isolation source. The keyword "algae" also includes *Cyanobacteria*. See Supplementary Data 2 for keyword definitions and original metadata. **c** Genome size (color) and estimated genome size (gray) calculated using estimated completeness scores from CheckM[83] ('lineage_wf'). Source data are provided as a Source Data file. **d** Species clusters with genetic potential for phototrophy (cyan markers); see Fig. 5b.

clustering together in the uncharacterized genus-level clade "Palsa-881" here renamed *Acaudatibacter* gen. nov. ('tailless bacterium') (Fig. 2b) (SeqCode taxonomic descriptions in Supplementary Notes 3 and 4). When also including *Acaudatibacter* genomes subset from the dataset of Rodríguez-Gijón et al.[37], which we refer to as the "extended dataset" (see "Methods"; Supplementary Data 1b), the number of *Acaudatibacter* species clusters lacking these genes increased to ten, comprising 32 MAGs from diverse sequencing projects (Supplementary Figs. S7a, b and S8). Given the large number of independently sampled and assembled *Acaudatibacter* genomes specifically lacking these genes, and that motility genes are normally spread out over the chromosome, the observed gene absence patterns are unlikely to stem from assembly artifacts. Moreover, these ten species formed a monophyletic group within *Acaudatibacter* (100% ultrafast bootstrap support [ufBS]), while the remaining two *Acaudatibacter* species, which have motility genes, *Ac.* sp. SZAS_AMP-5 and *Ac.* sp. CP_BM_RX_R9_33, were deep-branching (Supplementary Fig. S9). Thus, *P. immobile* and most members (10/12) of the new genus *Acaudatibacter* lack flagellar genes and are both robustly nested within clades where deeper branching lineages have flagellar genes. Together, this strongly suggests that flagellar motility has been lost in at least two separate *Caulobacteraceae* lineages. It also shows that the isolate *P. immobile* E^T is immotile due to a complete lack of flagellar genes, in contrast to *P. hankyongense* and *P. soli*, which despite being described as immotile[38,39] have flagellar genes (Fig. 2b and Supplementary Data 5b, c).

*C. crescentus* also produces polarly localized type IV pili (T4P), holdfast adhesin, and the prostheca (stalk) in a cell cycle-dependent manner (Fig. 2a). We found that *P. immobile* lacked genes for both holdfast and T4P, with the latter otherwise being highly conserved across *Caulobacteraceae* (Fig. 2b and Supplementary Data 5e, f). Similarly, the *Acaudatibacter* species lacking flagellar genes also lacked holdfast genes, in contrast to other members of the genus (Fig. 2b, Supplementary Fig. S8, and Supplementary Data 5e, f). Notably, since the genetic basis for the prostheca remains poorly defined, we could not robustly predict the presence of prosthecae among the species in our dataset (Supplementary Data 5i).

To elucidate how the cell developmental program has evolved to accommodate the absence of flagella and polar adhesion pathways, we searched for orthologs of *C. crescentus* developmental regulators. Remarkably, the genomes lacking flagellar and holdfast genes also lacked orthologs of the same 26 regulators of dimorphic development otherwise largely conserved across *Caulobacteraceae*, despite representing separate lineages (Fig. 2b, c, Supplementary Figs. S8 and S9, and Supplementary Data 5d). Moreover, except for two proteins (SpbR[40] and CpaM[41]), these regulators have all been associated with the signaling molecule cyclic diguanylate (c-di-GMP), a conserved bacterial regulator of motile–sessile transitions that in *C. crescentus* drives cell differentiation and orchestrates the asymmetric generation of motile offspring (Fig. 2a)[42]. This included producers and degraders of c-di-GMP (*dgcAB*, *pleD*, and *pdeA*, respectively), c-di-GMP-binding effectors (*popA*, *shkA*, *dgrAB*, *tipF*), as well as regulators closely upstream (*cckN*, *fssAB*, *hmrABCX*, *mopJ*), or downstream (*cpdR*, *rcdA*, *rpoN*, *shpA*, *shkA*, *spmY*, *staR*) of c-di-GMP (Fig. 2c). Despite the absence of this large set of c-di-GMP-associated genes, the genomes lacking flagellar and holdfast genes still encoded multiple enzymes with canonical c-di-GMP biosynthesis and/or degradation domains (Supplementary Fig. S10). Together, this suggests that these species still maintain c-di-GMP metabolism, but that they might use it for regulating other functions than their core developmental programs. Notably, in addition to the roughly 80–100 characterized cell morphology and development genes, we found 25 uncharacterized genes specifically absent from both non-flagellated *Caulobacteraceae* lineages (Supplementary Fig. S11 and Supplementary Data 6). These genes encode putative signaling proteins, transcriptional regulators, enzymes, and hypothetical proteins, and are promising candidate genes for potential flagellar motility and/or cellular dimorphism factors.

Given the absence of genes involved in morphological asymmetry among non-flagellated *Caulobacteraceae* species, we wondered if they also lack reproductive asymmetry, i.e., the production of daughter cells with different reproductive capabilities—a hallmark of the dimorphic lifecycle[43]. Indeed, when tracking two subsequent cell division cycles of *P. immobile* by time-lapse microscopy, we found that most daughter cell siblings descending from the first division cycle divided synchronously during the second division cycle (Fig. 3a–c and Supplementary Movie 1), similarly to the monomorphic bacterium *Escherichia coli* (Fig. 3a–c and Supplementary Movie 2). In contrast, in *C. crescentus*, one daughter cell consistently divided later than the other daughter cell, as expected (Fig. 3a–c and Supplementary Movie 3). Moreover, while *C. crescentus* exhibited a bimodal distribution of generation times among individual cells, consistent with the generation time difference between swarmer and stalked cells, monomodal distributions were instead observed for *P. immobile* and *E. coli* (Fig. 3d). Lastly, we also found that the cell division site was symmetrically positioned mid-cell in *P. immobile*, mirroring *E. coli*, but contrasting *C. crescentus*, in which it is asymmetrically localized closer to the swarmer cell pole[44] (Fig. 3e). These results demonstrate that *P. immobile* produces symmetric daughter cells, both regarding size and cell division timing.

Taken together, we show that genes associated with the canonical dimorphic lifecycle of *C. crescentus* are widespread across *Caulobacteraceae*, suggesting similar lifecycles among most members. However, our results indicate that *P. immobile* and a large subclade of *Acaudatibacter* have lost flagella and polar adhesins, along with specific cell developmental genes, suggesting convergent losses of dimorphic lifecycles. Moreover, we verified that *P. immobile*—the only cultured representative of these bacteria—reproduces symmetrically, in support of it having a monomorphic lifecycle.

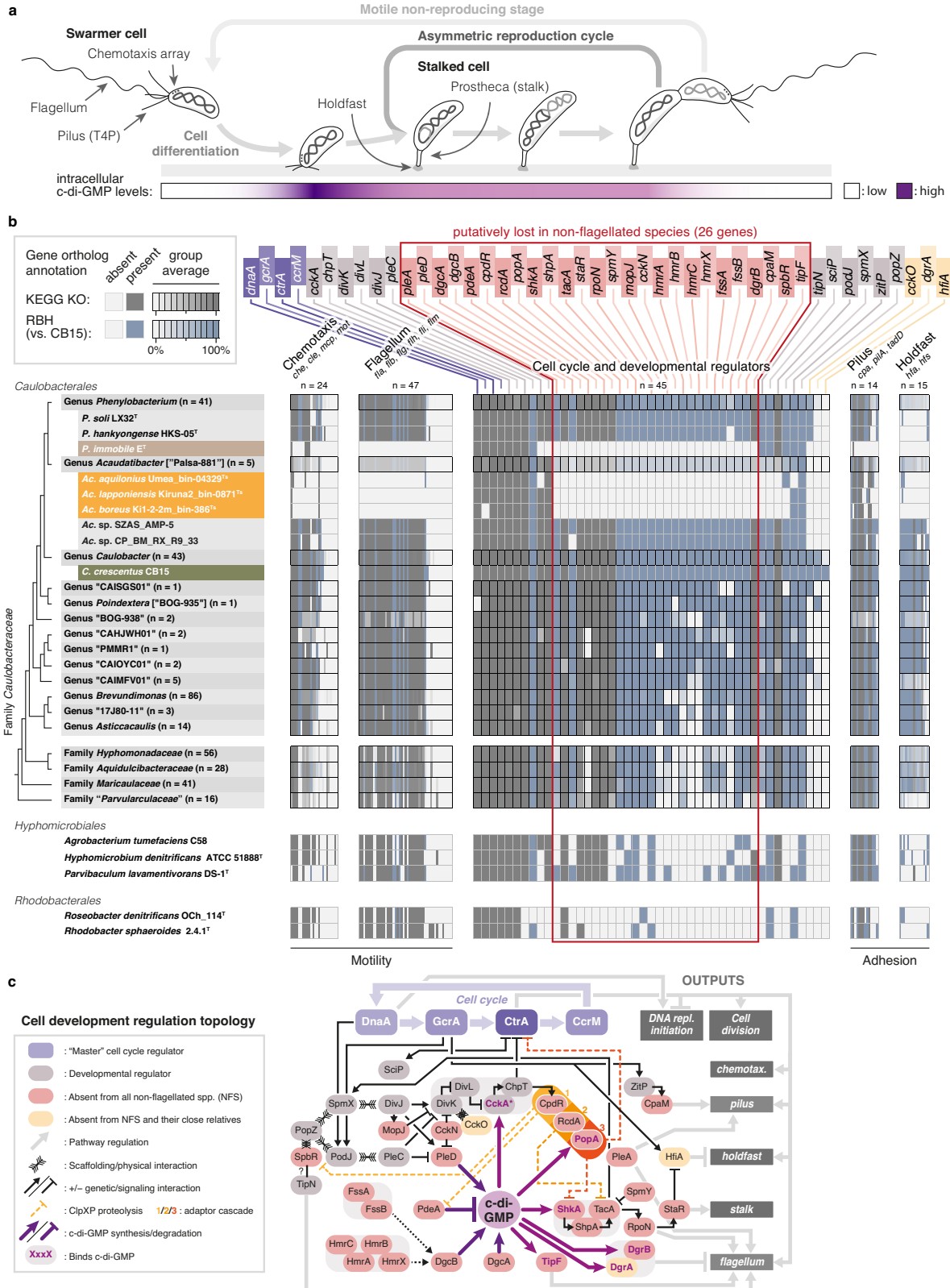

**Most *Caulobacterales* members lack genetic potential for S-layers and crescentin-mediated cell curvature**

Other well-studied[5] features of *C. crescentus* are its S-layer[45], a cell envelope layer assembled from the protein RsaA, and crescentin[46] (CreS), an intermediate filament-like cytoskeletal protein that generates cell curvature (Fig. 4a). We found that these two features were not particularly common among *Caulobacteraceae* (7.8% and 23.8% of species,

respectively) and that they were absent from other *Caulobacterales* families (Fig. 4b and Supplementary Data 5g, h). Specifically, the S-layer genes *rsaADEF* were limited to the environmentally widespread (see Fig. 1a, b) *C. crescentus–segnis* clade and disparate *Phenylobacterium* species, while *creS* was present among most *Caulobacter* and a minority of *Brevundimonas* species, with only singular hits found in a genome each of *Acaudatibacter*, "CAIMFV01", and *Phenylobacterium* (Fig. 4b).

**Fig. 2 | Convergent loss of cellular dimorphic traits within *Caulobacteraceae*.**
**a** Lifecycle of *Caulobacter crescentus*, showcasing the obligate motile-to-sessile (swarmer-to-stalked) cell differentiation that precedes reproduction, as well as the obligate generation of motile offspring through asymmetric cell division. Schematic representation of cellular levels of the intracellular signaling molecule cyclic diguanylate (c-di-GMP) is indicated in purple in the lower bar, highlighting the surge in c-di-GMP levels that drives cell differentiation[126]. **b** Presence and absence of genes involved in chemotaxis (*mcp* and genes with *che*, *cle*, and *mot* names), flagellar motility (genes with *fla*, *flb*, *flg*, *flh*, *fli*, and *flm* names), type IV tight-adherence pilus (T4P; *pilA*, *tadD*, and genes with *cpa* names), holdfast (genes with *hfa* and *hfs* names), and cell developmental regulation, as annotated by KEGG KOs (dark gray; from eggNOG-mapper[95]) and using the reciprocal best blast hit (RBH) algorithm against the *C. crescentus* CB15 proteome (blue). The total numbers of genes investigated for each functional category are indicated (*n*). For genera and families,

the average presence and absence is shown with linear color gradients. Five outgroup *Alphaproteobacteria* are included to highlight the broader conservation of genes. The labels of the cell developmental genes have been colored as follows. Purple: master cell cycle regulators; red: absent from both lineages lacking flagellar genes; yellow: absent from both lineages lacking flagellar genes and their close relatives; and gray: remaining regulators. See Supplementary Data 4 and 5a–f for the full dataset, including gene names and functional annotations. **c** Simplified schematic summary of *C. crescentus* cell development regulation, showcasing positive and negative regulatory interactions. Developmental regulators are color-coded as in (**b**). Direct binders of cyclic diguanylate (c-di-GMP) are highlighted with purple text. CckA is displayed in its phosphatase state (*). For simplicity, temporal aspects are disregarded, and most of the specific genes controlled by CtrA, CcrM, DnaA, and GcrA are not indicated. Based on refs. 7,40–42,127–144.

The limited distribution of crescentin prompted us to investigate its evolutionary origin. For this, we inferred a phylogeny of putative crescentin homologs (Fig. 4c and Supplementary Figs. S12a and S13). The resulting tree included phage tail tape-measure proteins (TMPs) from *Alpha-*, *Beta-*, and *Gammaproteobacteria*, and a fully-supported clade (100% ufBS) of sequences from *Caulobacteraceae* and the alphaproteobacterial order *Hyphomicrobiales* (synonym *Rhizobiales*), referred to here as 'crescentin-like', which included all proteins with the crescentin Pfam[47] domain (PF19220) (Supplementary Fig. S12b). TMPs are structural proteins that form extended coiled coils inside phage tails to regulate their length[48], reminiscent of the extended coiled coils of crescentin[49]. Crescentin-like proteins clustered with alphaproteobacterial TMPs, in lieu of most beta-/gammaproteobacterial TMPs, but with low support (81% ufBS) (Fig. 4c). While the simple repetitive primary sequences of coiled-coil proteins make them challenging to analyze phylogenetically[46], these results could hint at a potential phage ancestry of crescentin.

Apart from *Caulobacteraceae* sequences, which formed a distinct clade (100% ufBS), remaining crescentin-like proteins came from 13 bacterial families of the alphaproteobacterial order *Hyphomicrobiales* (Fig. 4d, Supplementary Fig. S13). Strikingly, the *Hyphomicrobiales* species included isolates previously shown to have curved morphologies (Fig. 4d, e), such as *Methylocystis parva*[12], *Methylocystis* sp. Rockwell[50], *Roseiarcus fermentans*[51], and *Chelatococcus reniformis*[52], whose crescentin-like protein is possibly deep-branching (90% ufBS) (Fig. 4c). To experimentally test if *Hyphomicrobiales* crescentin homologs can induce cell curvature, we complemented a non-curved ΔcreS mutant of *C. crescentus* with the *creS* homolog of *Ch. reniformis*. Indeed, *creS_Ch.reniformis* expression was sufficient to restore cell curvature in the ΔcreS mutant (Fig. 4f and Supplementary Fig. S14). Expression progressively generated left-handed cell spiraling upon longer incubation−a phenotype characteristic of *creS* overexpression[53]−confirming the presence of curvature-inducing crescentin outside of *Caulobacteraceae*. Together, our results reveal a restricted phylogenetic distribution of crescentin-like proteins across bacteria, being limited to *Hyphomicrobiales* and *Caulobacteraceae* (Fig. 4c).

### Phototrophy is widespread among *Caulobacterales*

In contrast to the unpigmented colonies of *C. crescentus*, some *Caulobacteraceae* strains produce carotenoid pigments ranging from yellow to reddish orange (Supplementary Data 7 and 8b). When analyzing the distribution of carotenoid biosynthesis genes, we found that many species possessed *crtCDF* (Supplementary Data 8c)−genes used by photosynthetic bacteria to make carotenoids aiding light capture and electron transfer, and acting as photodamage protectants[54,55]. This observation, together with recent reports of phototrophic potential among individual *Caulobacterales* species[56–59], prompted us to systematically investigate the distribution of phototrophy genes across *Caulobacterales*.

Unexpectedly, we found type II anoxygenic photosynthesis genes in 10% (35/347) of *Caulobacterales* species representative genomes. This included reaction center (RC) subunits L, M, and H (*pufLMH*), light-harvesting complex I (LH1) α/β subunits (*pufAB*), chlorophyll transporters (*pucC*), and numerous bacteriochlorophyll biosynthesis enzymes (*acsF* and *bch* genes) (Fig. 5a,b and Supplementary Data 8d–g). Phototrophy genes always co-occurred with a combination of the aforementioned carotenoid genes *crtCDF* (Fig. 5b), which were otherwise absent among *Caulobacterales* (Supplementary Data 8c–g). Furthermore, photosynthesis genes were generally organized in a gene cluster (Fig. 5c and Supplementary Fig. S15), as is typical for *Pseudomonadota*[60].

We found phototrophic potential in members of all *Caulobacterales* families, totaling at least 16 genera (Figs. 1d and 5b and Supplementary Data 8). This mostly included uncultured species of previously uncharacterized lineages, but also characterized *Aquidulcibacter*, *Brevundimonas*, and *Oceanicaulis* (*Alkalicaulis*) species, such as the dark-pigmented *Brevundimonas* isolates *B. bacteroides* CB7[T], *B. subvibrioides* CB81[T], and *B. variabilis* CB17[T]. Absorbance scans of cell suspensions of these strains, when grown aerobically in ambient light, confirmed the production of *a*-type bacteriochlorophyll in *B. variabilis* CB17[T] (Fig. 5d). However, *B. bacteroides* CB7[T] and *B. subvibrioides* CB81[T] lacked detectable bacteriochlorophyll, suggesting that phototrophy was not induced for these species under the chosen growth conditions.

Notably, we also found phototrophy genes among the three putatively monomorphic *Acaudatibacter* species of our core dataset (Fig. 5b and Supplementary Fig. S16; see Fig. 2b). In addition to RC−LH1 supercomplex genes, the three *Acaudatibacter* species also encoded the α and β subunits (*pucAB*) of the auxiliary light-harvesting complex II (LH2) (Fig. 5a−c and Supplementary Data 8f, g). Unexpectedly, two of the species, *Ac. aquilonius* and *Ac. lapponiensis*, had genes for RuBisCO (*cbbLS*) and the complete Calvin-Benson-Bassham (CBB) cycle for carbon fixation (Fig. 5b and Supplementary Data 8h and 9a, b). This suggests that they might not only capture light energy primarily for ATP synthesis, as is most common for alphaproteobacterial phototrophs (photoheterotrophy)[61], but that they can also use light to fix carbon (Fig. 5a). In support of such photoautotrophic metabolism, *Acaudatibacter* CBB cycle genes were organized inside the photosynthesis gene cluster (PGC) (Fig. 5c), which also included accessory genes for red-type RuBisCO activase (*cbbX*) and XuBP phosphatase (*cbbY*). The genome of *Ac. boreus*, the third *Acaudatibacter* species, probably also encodes a complete CBB cycle, since we could identify orthologs for all enzymatic steps except for *cbbLS* (Supplementary Data 9a, b), which was likely incompletely assembled, since part of the *cbbL* sequence was detected on a contig edge (Fig. 5c). Notably, complete and near-complete CBB cycles were also found among two *Vitreimonas* phototrophs (Fig. 5b, Supplementary Data 8h and 9a, b). *Acaudatibacter* PGCs were highly syntenic to other *Caulobacteraceae* PGCs, with the *Acaudatibacter*-specific LH2 and carbon fixation genes

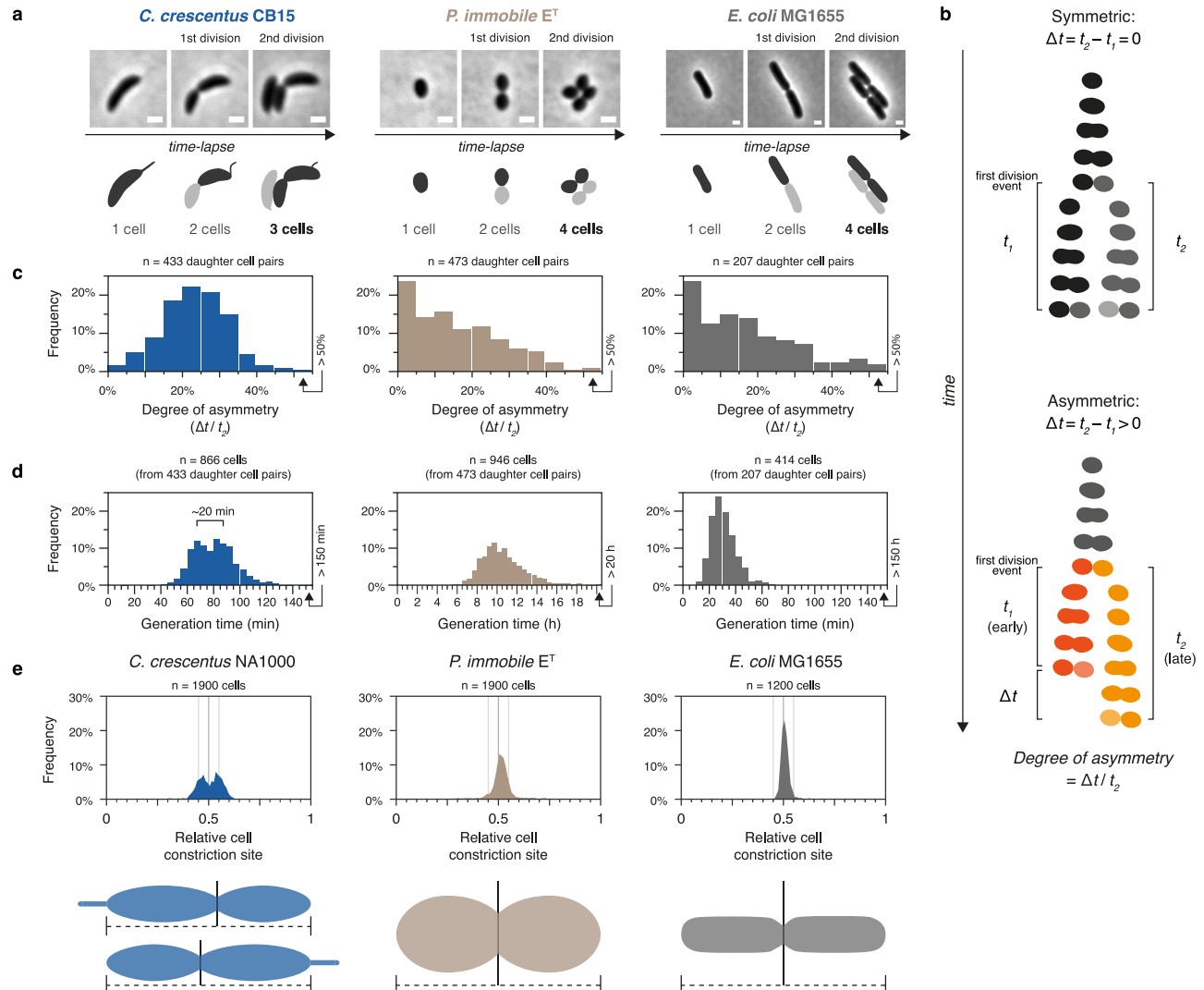

**Fig. 3 | *Phenylobacterium immobile* reproduces symmetrically. a** Micrographs from timelapses at the time of the first and second cell division events for *Caulobacter crescentus* CB15 (PYE, 30 °C), *P. immobile* E$^T$ (R2A, 30 °C), and *Escherichia coli* MG1655 (LB, 37 °C). Cell outline schematics highlight daughter cell lineages. Full timelapses are shown in Supplementary Movies 1–3. Scale bars: 1 μm. **b** Schematics of the quantification of degrees of replicative asymmetry from time-lapse microscopy. The daughter cells of an initial binary cell division event divide again after the times $t_1$ and $t_2$, respectively. For bacteria with replicative asymmetry among daughter cells, one daughter cell undergoes subsequent cell division at a later time point ($t_2$), than its sibling ($t_1$), i.e., $\Delta t = t_2 - t_1 > 0$. To compare species with different growth rates, we defined the *degree of asymmetry* as $\Delta t / t_2$, i.e., how much extra time the late-dividing daughter cells spends to divide, as a proportion of their total generation time. **c** Histograms of *degrees of asymmetry* calculated for daughter cell pairs as described in (**b**), from time-lapse experiments conducted as in (**a**) for *C. crescentus*, *P. immobile*, and *E. coli* in at least three biologically independent replicates (in total $n = 433$, $n = 473$, and $n = 207$ daughter cell pairs, respectively). The right-most bins represent *degrees of asymmetry* above 50%. Source data are provided as a Source Data file. **d** Histograms of generation times of individual cells, from daughter cell pairs analyzed in (**c**). For *C. crescentus* CB15, the approximate time difference between the bimodal peaks is highlighted. The right-most bins represent generation times above 150 min for *C. crescentus* and *E. coli*, or above 20 h for *P. immobile*. Source data are provided as a Source Data file. **e** Histograms of relative cell constriction site positions in *C. crescentus* NA1000 ($n = 1900$), *P. immobile* E$^T$ ($n = 1900$), and *E. coli* MG1655 ($n = 1200$), divided into 100 bins, quantified from snapshot images of cells grown in batch cultures to mid-exponential phase under growth conditions corresponding to those used for timelapses presented in (**a**). Vertical lines show mid-cell position ±5%. Schematic cell illustrations highlight that relative constriction site positions are not sorted by cell polarity. Source data are provided as a Source Data file.

being organized outside of the syntenic region (Fig. 5c and Supplementary Fig. S15), suggesting related origins of the PGCs despite photosystem complexity differences.

To investigate the evolutionary history of *Caulobacterales* phototrophy genes, we inferred phylogenies for the reaction center protein PufM and bacteriochlorophyll enzyme BchY. Aside from *Maricaulaceae* (M), *Caulobacterales* PufM orthologs clustered together (96% ufBS), with individual genera forming subclades, suggesting close relatedness of PufM across most *Caulobacterales* (Fig. 5e and Supplementary Fig. S17). The *Caulobacterales* clade was in turn affiliated (46% ufBS) with a clade comprising sequences from

*Maricaulaceae* (M) and *Sphingomonadales* (an alphaproteobacterial order) (Fig. 5e and Supplementary Fig. S17). The affiliation with *Sphingomonadales* was consistent and well-supported (97% ufBS) in another PufM tree (Supplementary Figs. S18a and S19), and matches previous phylogenetic analyses that included *B. subvibrioides*[62]. However, in our alternative PufM tree, *Hyphomonas* (H) and "*Parvularculaceae*" (P) sequences clustered outside the main *Caulobacterales* clade (Supplementary Figs. S18a and S19). Similar to PufM, in BchY phylogenies, most *Caulobacterales* sequences clustered with each other and with *Sphingomonadales*, although with larger numbers of *Caulobacterales* sequences branching outside with

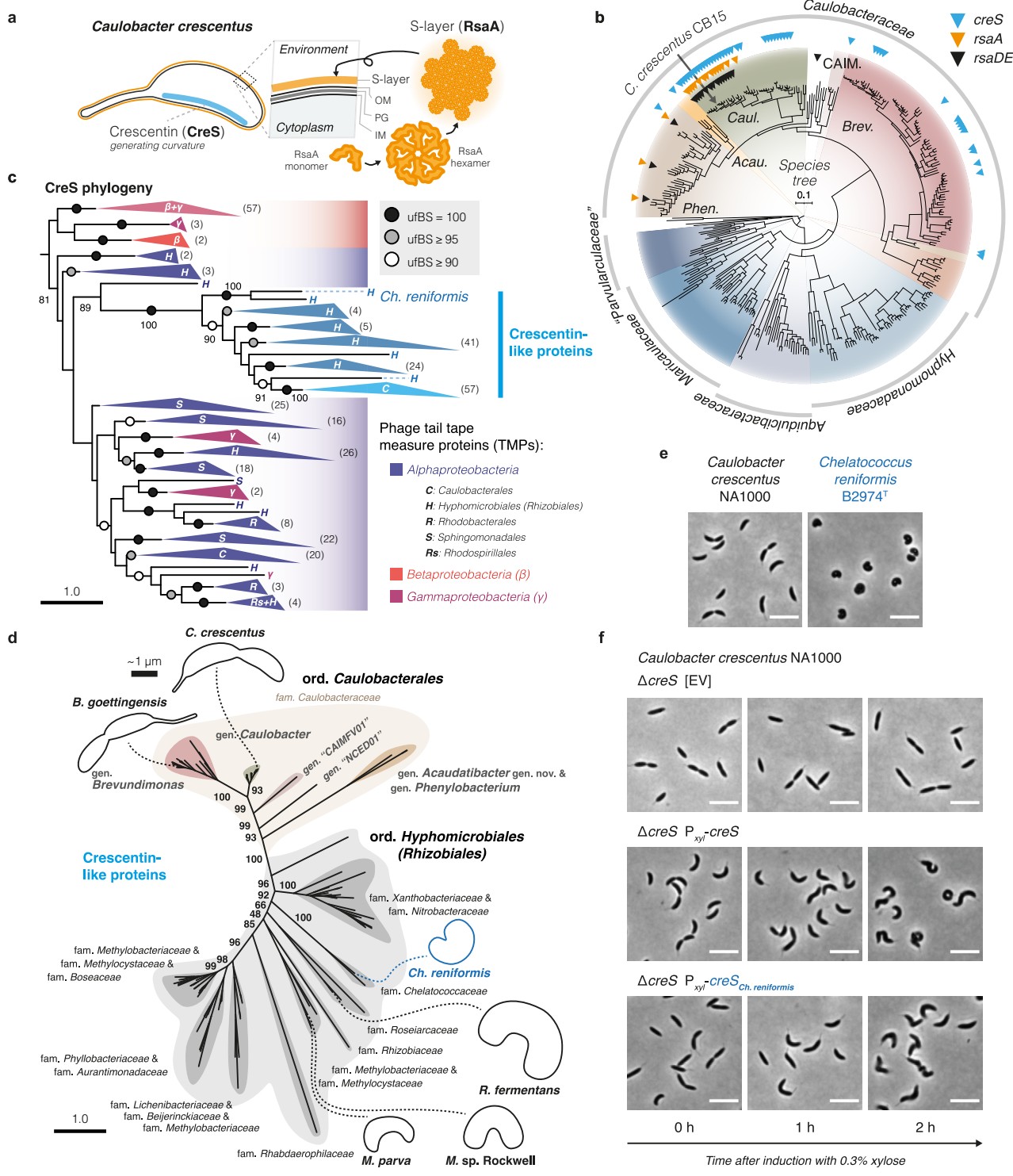

unsupported positions (Supplementary Figs. S18b, c and S20–S21). While we were unable to further resolve the topology of *Caulobacterales* PufM/BchY, the patchy but widespread representation of phylogenetically affiliated photosynthesis genes across all major *Caulobacterales* clades (Fig. 5b, e and Supplementary Data 8a–h), organized in largely syntenic gene clusters (Fig. 5c and Supplementary Fig. S15), could suggest that the ancestor of *Caulobacterales* was potentially a phototroph. It is also possible that horizontal transfer of the PGC has occurred between different *Caulobacterales* lineages, followed by vertical inheritance within individual clades, as horizontal transfers are more frequent between closely related lineages[63].

## *Acaudatibacter* species with photoautotrophic potential inhabit anoxic layers of stratified boreal lakes in high relative abundance

Given the unusual genetic potential for photoautotrophic metabolism and the lack of dimorphic lifecycle genes in *Ac. aquilonius*, *Ac. boreus*, and *Ac. lapponiensis*, we wondered if this *Acaudatibacter* lineage inhabits an ecological niche atypical for *Caulobacteraceae*. To this end, we analyzed a dataset of 24,050 prokaryotic species-cluster genome representatives that have been competitively mapped against 636 freshwater metagenomes (Supplementary Figs. S22a, b)[37]. In the dataset, *Ac. aquilonius*, *Ac. boreus*, *Ac. lapponiensis*, as well as a fourth species cluster (*Ac.* sp. 23796) not part of our initial core dataset

**Fig. 4 | Distribution and evolution of key *Caulobacter crescentus* morphogenic genes. a** Schematic illustration of the cell curvature-generating cytoskeletal protein crescentin (CreS) and S-layer cell envelope protein RsaA in *C. crescentus*. OM outer membrane, PG peptidoglycan, IM inner membrane. **b** Presence (triangles) of crescentin (*creS*; K18642; blue), S-layer (*rsaA*; K12544; orange), and S-layer secretion machinery (*rsaDEF*; K12533–K12535; black) across *Caulobacterales* genomes. The phylogeny shows the tree from Fig. 1a. *Phen.*, *Phenylobacterium*; *Acau.*, *Acaudatibacter*; *Caul.*, *Caulobacter*; CAIM., "CAIMFV01"; *Brev.*, *Brevundimonas*. See Supplementary Data 4 and 5g, h for the full dataset. **c** ML tree of crescentin (CreS) homologs inferred using IQ-TREE[109] with the Q.pfam + C60 + R9 model of evolution and 1000 ultrafast bootstraps (356 sequences with 902 amino acid alignment positions), and rooted using the major *Beta-/Gammaproteobacteria* clade. ufBS is indicated with circles and specified with numbers for key branches. Greek letters indicate *Pseudomonadota* (*Proteobacteria*) classes, and uppercase letters indicate alphaproteobacterial orders (see Supplementary Fig. S12a). The numbers of sequences per collapsed clade are shown in parentheses. Scale bars here and in (**d**) indicate the number of substitutions per site in the phylogenies. The full tree is presented in Supplementary Fig. S12a. **d** ML tree inferred separately for the subset of putative crescentin homologs that form a supported clade (100% ufBS) labeled "Crescentin-like proteins" in the tree shown in (**c**), using IQ-TREE[109] with the JTT + C60 + R5 model of evolution and 1000 ultrafast bootstraps (135 sequences with 407 amino acid alignment positions). ufBS is indicated with numbers for key branches. Schematic cell silhouettes are drawn roughly in scale (thick scale bar: ~1 μm) based on microscopy images presented in Supplementary Fig. S4 for *Brevundimonas goettingensis*, *C. crescentus*, and *Chelatococcus reniformis*, and from published material for remaining species (*Methylocystis parva*, *Methylocystis* sp. Rockwell, and *Roseiarcus fermentans*; references are listed in Supplementary Table S1). gen. genus, fam. family, ord. order. The full tree is presented in Supplementary Fig. S13. **e** Representative micrographs (from two independent experiments each) of *C. crescentus* NA1000 and *Ch. reniformis* B2974ᵀ from cultures in mid-exponential phase, grown in PYE at 30 °C and R2A at 28 °C, respectively. Scale bars: 5 μm. **f** Complementation of *C. crescentus* NA1000 Δ*creS* with native *creS* or the crescentin homolog from *Ch. reniformis* B2974ᵀ (*creS*$_{Ch.reniformis}$) expressed from the xylose-inducible P$_{xyl}$ promoter on the pBXMCS-4 vector. Representative micrographs (from one experiment) are shown for mid-exponential cultures before (0 h) and after the addition of 0.3% xylose. Scale bars: 5 μm. EV empty vector. See Supplementary Fig. S14 for corresponding experiments done in the wild-type NA1000 background.

(Supplementary Fig. S7), were among the most prevalent and relatively abundant *Caulobacteraceae* freshwater species (Supplementary Fig. S23).

The MAGs of these four *Acaudatibacter* species derive from short-read metagenomes collected together with water chemistry measurements along the water columns of stratified freshwater bodies[64]. Genomes of these species recruited reads from metagenomes from twelve boreal freshwater bodies from Northern Sweden, Finland, and Canada (Fig. 6 and Supplementary Figs. S22c and S24). In most cases, the *Acaudatibacter* species were most abundant right below the oxycline (Fig. 6c). This pattern contrasts with the predominantly aerobic lifestyles of *Caulobacteraceae*[11], but is typical for anoxygenic phototrophs that can be found in the illuminated upper parts of the hypolimnion, where they use reduced compounds found under anoxic conditions as electron donors[65]. Given these patterns, we suspect that *Ac.* sp. 23796 could also encode a PGC that was undetected due to genome incompleteness (51.0–92.5% completeness of species cluster members) (Supplementary Fig. S7a). In some sample series, *Ac. aquilonius* and *Ac.* sp. 23796 were most abundant in the oxic upper layers (Fig. 6c), suggesting that these species are not obligate anaerobes. Indeed, consistent with a facultatively aerobic lifestyle, all four *Acaudatibacter* species representatives have the genetic potential for aerobic respiration (Fig. 6a, Supplementary Fig. S16, and Supplementary Data 8i, j). Moreover, the non-homogeneous water column distribution of the *Acaudatibacter* species suggests that they actively position themselves, despite lacking flagellar genes, perhaps by regulating buoyancy through pilus-mediated cellular aggregation, as previously reported for other anoxygenic phototrophs[65].

## Discussion

Knowledge about the ecology and evolutionary context of many well-characterized bacterial model organisms is surprisingly limited, restricting our ability to apply insights from experimental studies to the diversity of environmental bacteria. In this study, we provide a comprehensive portrait of the evolution and environmental distribution of the model organism *C. crescentus* and members of its alphaproteobacterial family *Caulobacteraceae* and order *Caulobacterales*. Our work sheds light on the biology of hundreds of previously uncharacterized species and uncovers unexpected diversity within this bacterial order.

*C. crescentus* has long served as a model for bacterial cell biology[6] and has significantly advanced our understanding of bacterial development and morphogenesis. We found that some of the characteristic morphological features of *C. crescentus*, including its S-layer and crescentin-mediated curved shape, are not common characteristics across *Caulobacteraceae* and are even absent from other *Caulobacterales* families (Fig. 4b). However, we identified and experimentally confirmed the presence of crescentin-like proteins in the sister order *Hyphomicrobiales* (Fig. 4d–f). Crescentin-mediated cell curvature is thought to improve flagellar motility[5,46] and pilus-mediated surface colonization[66] in *C. crescentus*, but interestingly, two of the curved crescentin-encoding *Hyphomicrobiales* species (*Ch. reniformis* and *R. fermentans*) lack genes for flagella and pili (Supplementary Fig. S25). Thus, crescentin-mediated curvature might provide yet another selective advantage in these species. Additionally, our phylogenetic analyses could indicate that crescentin has evolved from a prophage protein, either initially in *Hyphomicrobiales*, later being horizontally transferred to *Caulobacteraceae*, or in their common ancestor, with extensive loss in the *Caulobacterales* lineages. While this finding requires further investigation, it would explain the taxonomically limited distribution of crescentin among bacteria (Fig. 4c).

While our data revealed a limited distribution of crescentin and the S-layer, most of the numerous genes important for the motile–sessile dimorphic lifecycle of *C. crescentus* are present across both *Caulobacteraceae* and *Caulobacterales* (Fig. 2b and Supplementary Data 5a–f). However, we identified eleven species from two evolutionarily independent lineages that appear to have lost essentially all genes known to mediate flagellar motility, chemotaxis, and polar holdfast adhesion, alongside over 20 developmental regulators. This includes ten species clusters of the herein described genus *Acaudatibacter* gen. nov., and *P. immobile* (genus *Phenylobacterium*)—the only cultured representative—which we experimentally verified to lack dimorphic reproductive asymmetry (Fig. 3). Thus, our findings uncover the first clear example of a lineage (*P. immobile*) in which an obligate dimorphic lifecycle has reverted to a simpler monomorphic lifecycle. Moreover, the remarkably similar large absence of genes among the *Acaudatibacter* species compared to *P. immobile* strongly suggests convergent loss of dimorphism and reveals core modules specifically used for establishing bacterial dimorphism. The nearly 100 morphogenesis and developmental genes absent from these lineages are distributed across the chromosome, rather than being part of one or a few gene clusters, indicating that their combined absence is the result of numerous mutations driven by considerable evolutionary pressure. Dimorphism is common to the alphaproteobacterial *Caulobacterales*–*Hyphomicrobiales*–*Rhodobacterales*–*Sphingomonadales* superclade, estimated to have diverged ~1.5 billion years ago[67,68], and analogous dimorphic lifecycles are found in diverse bacteria[69–73], yet it remains unclear precisely what has driven the evolution of these complex bacterial lifecycles. Understanding the evolutionary processes that revert dimorphic lineages into monomorphs will help

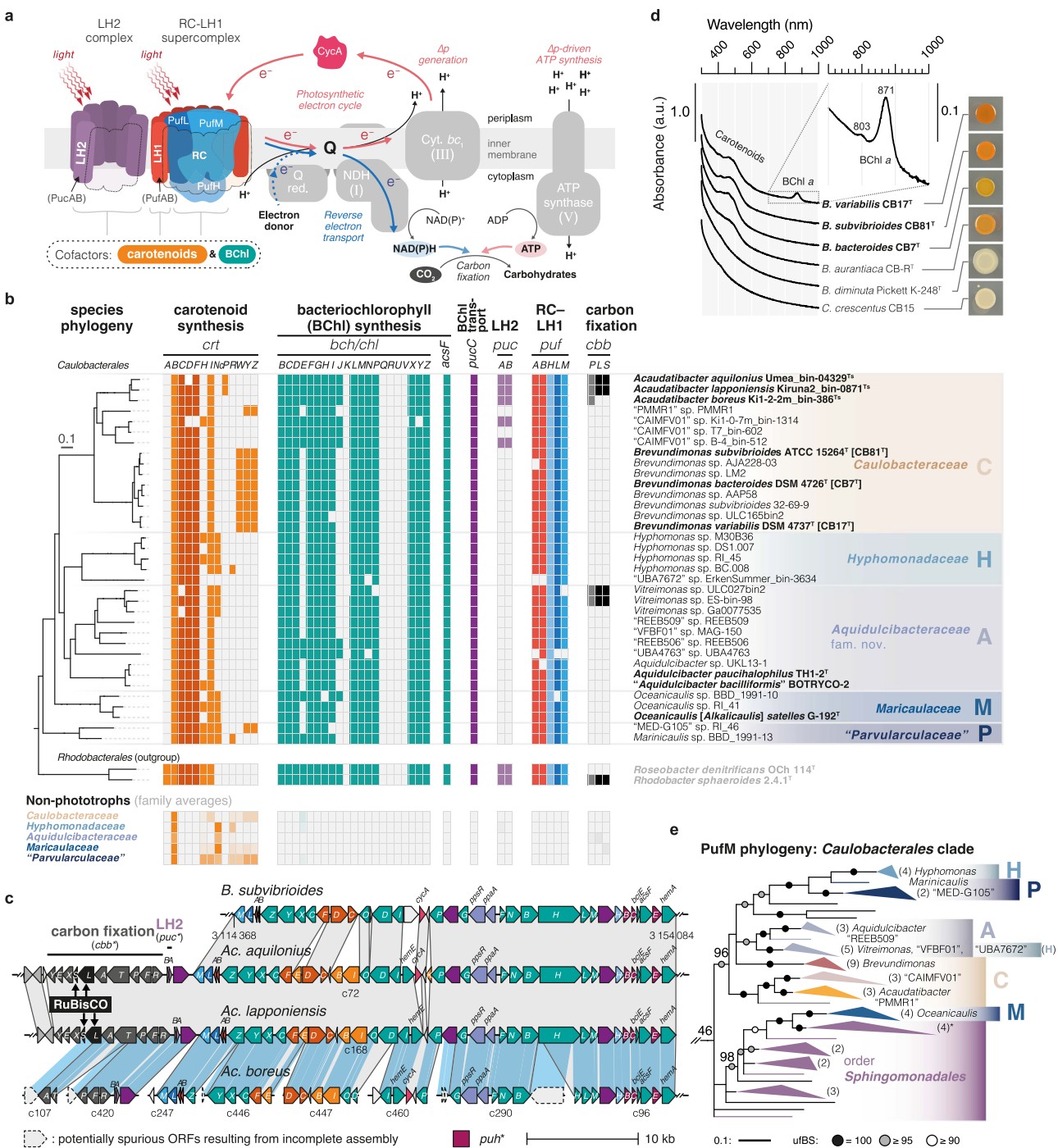

clarify the selective advantages and disadvantages of bacterial dimorphism.

It is noteworthy that virtually all regulatory developmental modules that were found to be lost in *P. immobile* and *Acaudatibacter* species through convergent evolution are intimately linked to the signaling molecule c-di-GMP (Fig. 2c), a key driver of motile–sessile transitions in bacteria, including the dimorphic lifecycle of *C. crescentus*[42]. Yet, we found that these lineages still have multiple genes encoding canonical c-di-GMP enzymes (Supplementary Fig. S10), indicating extensive restructuring of the c-di-GMP signaling network to orchestrate other functions than the dimorphic lifecycle. Precisely how this large rewiring of cellular signaling pathways has been accomplished is an exciting open question that will be worthwhile to address by exploiting the experimental tractability of *P. immobile* and its close phylogenetic relationship with the well-established model organism *C. crescentus*.

The new genus *Acaudatibacter* not only stands out regarding its inferred lifecycle, but also its metabolic potential. In three of the putatively monomorphic *Acaudatibacter* species, we have identified photosynthesis gene clusters containing CBB cycle genes supporting carbon fixation (Fig. 5c). Together with our finding that these species inhabit both oxic and anoxic layers of stratified freshwater bodies in high relative abundance (Fig. 6 and Supplementary Fig. S23), this suggests possible roles for these bacteria as primary producers—a previously unrecognized role for *Caulobacterales*. While the *Acaudatibacter* species stood out with their potential for full photosynthesis, including carbon fixation (Supplementary Fig. S15 and Supplementary Data 8 and 9), we found that phototrophic potential is surprisingly widespread in *Caulobacterales* (~10% of species across all major clades), most of which remain uncultured (Fig. 1b, d). This discovery is in line with recent reports uncovering phototrophic potential in other

**Fig. 5 | Phototrophy is widespread among *Caulobacterales*. a** Schematic illustration of a type II anoxygenic photosystem. LH2 light-harvesting complex II, RC–LH1 reaction center-light harvesting complex I, BChl bacteriochlorophyll, Q ubiquinone/ubiquinol electron carrier, Q red. quinone reductase, NDH NADH dehydrogenase, Cyt. $bc_1$ cytochrome $bc_1$, $\Delta p$ proton gradient. Roman numerals indicate orthologs of mitochondrial respiratory chain complexes. **b** Presence (color) and absence (light gray) of type II anoxygenic photosynthesis genes (KEGG orthologs) among putative phototrophs (top panel) and all remaining genomes averaged over each *Caulobacterales* family (bottom panel). Species phylogeny shows the tree from Fig. 1a pruned to include only relevant species. See Supplementary Data 4 and 8 for the full dataset. **c** Synteny of photosynthesis gene clusters (PGCs) from *Acaudatibacter* gen. nov. phototrophs compared to *B. subvibrioides* CB81[T], showing that LH2 and carbon fixation genes are encoded inside the *Acaudatibacter* PGCs. Gray boxes link orthologs. Blue boxes highlight sequences aligned using progressiveMauve[105] to map the contigs of the incompletely assembled PGC of *Ac. boreus* against the fully assembled PGC of *Ac. lapponiensis*. Contigs are marked with "c" followed by the contig number. Color coding corresponds to panel (**b**). **d** Colony spots and absorbance scans of colony material from putative phototrophs (bold), as well as non-phototrophic pigmented (CB-R[T]) and unpigmented (Pickett K-248[T] and CB15) control strains after 6 days on PYE agar at 23 °C in ambient light. The insert highlights the peaks at 871 nm (major) and 803 nm (minor), characteristic of bacteriochlorophyll $a$[145], for *B. variabilis* CB17[T]. Source data are provided as a Source Data file. **e** The *Caulobacterales*-containing clade of an ML tree of photosynthetic reaction center subunit PufM. The phylogeny was inferred using IQ-TREE[109] with the WAG + C60 + R9 model of evolution and 1000 ultrafast bootstraps (278 sequences with 295 amino acid positions). ufBS is indicated with shaded circles and specified with numbers for key branches. The clade comprising *Caulobacterales* phototrophs (ufBS = 46%) was pruned from the tree, and unsupported branches (ufBS < 80%) were deleted. The number of sequences per collapsed subclade is shown in parentheses. Uppercase letters represent *Caulobacterales* families; see panel (**b**). The asterisk indicates a clade in which a second copy of PufM from *Hyphomonas* sp. DS1.007 clusters with three *Sphingomonadales* sequences. The scale bar indicates the number of substitutions per site. The full tree is presented in Supplementary Fig. S17.

groups of environmental bacteria through cultivation-independent comparative genomics[74]. It is noteworthy that the putative *Caulobacterales* phototrophs include species with various morphologies and lifecycles, comprising both dimorphic and putatively monomorphic species, with or without prosthecae, even including members of prosthecate budding lineages (Figs. 1a, d and 5b). This raises questions as to how photosystem assembly is coordinated with the dimorphic developmental program. For example, the prosthecae (stalks), which are extensions of the cell envelope that increase cellular membrane surface area, might participate in light capture, similar to the intracellular membrane invaginations produced by classical phototrophic bacteria[61]. Moreover, our finding that *Caulobacterales* photosynthesis genes mostly form one supported clade (Fig. 5e and Supplementary Fig. S18) suggests that phototrophy could represent an ancestral *Caulobacterales* trait maintained in a minority of extant members. Anoxygenic type II photosynthesis has an ancient and complex history in *Alphaproteobacteria*, which remains unresolved[62]. Incorporating the rich diversity of *Caulobacterales* phototrophs into phylogenetic investigations is expected to shed new light on the evolution of bacterial photosynthesis.

## Methods

### Bacterial strains and growth conditions

Bacterial strains listed in Supplementary Table S2 were routinely grown using peptone yeast extract (PYE; 0.2% ($w/v$) Bacto peptone, 0.1% ($w/v$) Bacto yeast extract, 1 mM $MgSO_4$, 0.5 mM $CaCl_2$) or in R2A[75] (at pH 7.2, or adjusted to pH 6.0 using 37% HCl before autoclaving), or LB (Carl Roth), as specified. When necessary, media were solidified with 1.5% ($w/v$) Difco agar (BD). For plasmid selection and maintenance, media were supplemented with gentamicin (15 μg/ml for LB broth, 20 μg/ml for LB agar, 0.625 μg/ml for PYE broth, 5 μg/ml for PYE agar). Liquid cultures were incubated in flat-bottomed E-flasks, with 200 rpm orbital shaking. Unless specified, cultures were incubated at 30 °C. Strain stocks were preserved in 10% DMSO at −80 °C.

### Strain construction

Oligonucleotides (synthesized at Eurofins Genomics) are listed in Supplementary Table S3. The *creS* gene from *C. crescentus* NA1000 (KJ1) and the *creS* homolog (WP_188608507.1; IEW34_RS07480) from *Ch. reniformis* B2974[T] (KJ1171) were amplified from genomic DNA using primer pairs OJH27/OJH28 and OJH29/OJH30, respectively. The *creS* fragments were then inserted by Gibson assembly[76] into the pBXMCS-4 vector[77] linearized by PCR using primers OJH25 and OJH26, creating plasmids pBXMCS-4-$P_{xyl}$-*creS* and pBXMCS-4-$P_{xyl}$-*creS*$_{Ch.reniformis}$. The resulting plasmids and the original pBXMCS-4 empty vector (EV) were transformed into *C. crescentus* NA1000 (KJ1) and *C. crescentus* NA1000 Δ*creS* (KJ1179) by electroporation.

### Microscopy and image analysis

Cells were spotted onto 1% ($w/v$) agarose pads prepared with appropriate growth media, and phase-contrast images were acquired using an ECLIPSE Ti inverted research microscope (Nikon) equipped with a Plan Apo λ 100× Oil Ph3 DM (1.45 NA) objective (Nikon) and a Zyla sCMOS camera (Andor). For time-lapse imaging, cells were spotted onto growth medium agarose pads prepared inside Gene Frames (65 μl, 1.5 × 1.6 cm; Thermo Fisher) on glass slides, sealed with a glass coverslip, incubated at the specified temperatures, and illuminated only during imaging (a few seconds). For timelapses of *C. crescentus*, *E. coli*, and *P. immobile*, imaging intervals of 30 s, 20 s, and 5 min were used, respectively. Relative constriction site position was determined using the plugin MicrobeJ (v5.13l) in Fiji (ImageJ; v2.1.0/1.53c). Differences in replication timing for daughter cell pairs were calculated from manually tracked timelapses, as outlined in Fig. 3b, by following daughter cells stemming from the first cell division events after being spotted onto the agarose. Time-lapse images (Fig. 3a) and videos (Supplementary Movies 1–3) were subjected to linear stack alignment with SIFT using default settings in Fiji (ImageJ; v1.54 f).

### Pigment analyses

For agar spots, material from PYE agar colonies incubated at 23 °C for 6 days was suspended in PYE and diluted to an $OD_{600}$ of 0.1, before 10 μl was spotted onto a PYE agar plate. The resulting growth patches were photographed after 6 days at 23 °C in ambient light. For in vivo absorbance scans, PYE plate cultures were grown at 23 °C for 6 days in ambient light before cells were suspended in 10 mM Tris-HCl buffer (pH 7.8). To reduce light scattering, the cell suspension was mixed with glycerol in a 3:7 (suspension:glycerol) ratio, before one-nanometer absorbance spectrum scans (200–1000 nm) were recorded using a Spark multimode microplate reader (TECAN) at 23 °C.

### Isolation and sequencing of *Caulobacter* strains

A pre-enrichment of Lake Erken (Norrtälje, Sweden) microorganisms was established in dissolved organic matter produced by *Microcystis aeruginosa* (cyanobacterium) in a previous study[78] on February 28th, 2019. The pre-enrichment was incubated at 20 °C in 12-h light–dark cycles. The pre-enrichment was then diluted a thousand-fold to 1000 cells/ml on April 29th, 2019, for a second round of enrichment with DOM from *M. aeruginosa*, and incubated under the same conditions. On October 7th, 2021, the presence of prosthecate dimorphic bacteria in the second batch enrichment culture was confirmed by microscopy, and culture liquid was streaked on PYE and incubated in ambient light at 23 °C for 6 days. Seven colonies were re-streaked and incubated for another 6 days, before a well-isolated single colony from each of the resulting seven strains was inoculated into 10 ml PYE broth and incubated at 23 °C with 200 rpm shaking. After three days,

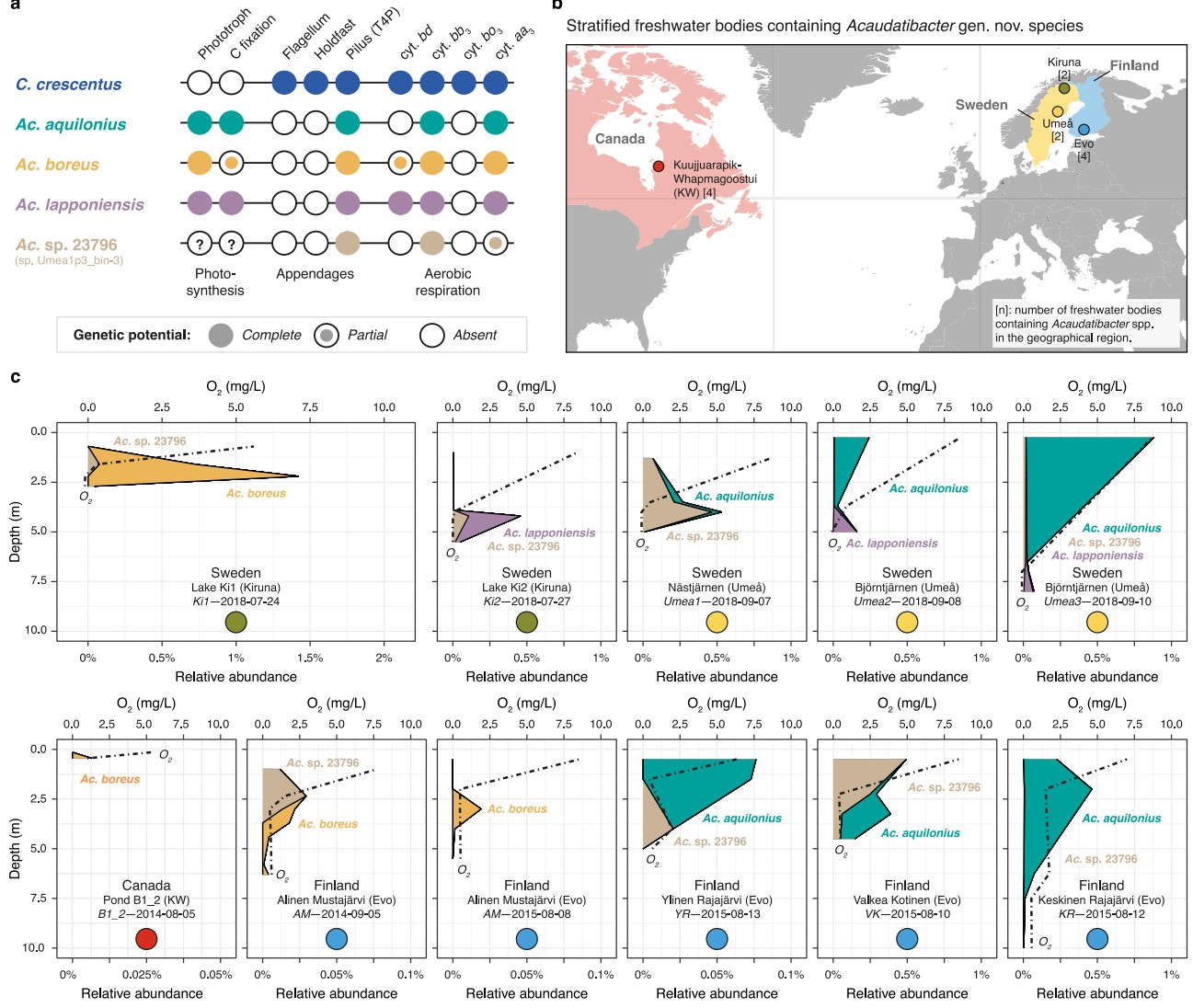

**Fig. 6 | Environmental distribution and relative abundance of *Acaudatibacter* gen. nov. species with photoautotrophic potential. a** Overview of the genetic potential of *Acaudatibacter* species present in stratified freshwater bodies (based on Figs. 2b and 5b, Supplementary Figs. S8 and S16, and Supplementary Data 9). The genetic potential of *C. crescentus* is simply included as a point of reference. Question marks emphasize that, given their genome incompleteness (Supplementary Fig. S7a), the genome assemblies of *Ac.* sp. 23796 could lack photosynthetic potential due to the photosynthesis gene cluster not being assembled. T4P type IV pilus, cyt. cytochrome. **b** Geographical regions with stratified freshwater bodies containing putatively photoautotrophic *Acaudatibacter* species in

metagenomes from Buck et al.[64] Numbers within square brackets denote the number of *Acaudatibacter*-containing freshwater bodies in the region (see Supplementary Fig. S22c). **c** Relative abundance of *Acaudatibacter* species among metagenomes sampled across the water column of stratified freshwater bodies. Dashed lines show $O_2$ concentrations measured during sampling. For each sampling series, the name of the freshwater body, geographical region (parentheses), lake code (italics), and sampling date are indicated. Colored circles represent geographical regions shown in (**b**). Additional Canadian freshwater bodies with *Acaudatibacter* presence, but lacking depth profiles, are presented in Supplementary Fig. S24. Source data are provided as a Source Data file.

cultures were preserved in 10% DMSO at −80 °C. Out of seven strains, three were yellow-pigmented rosette-forming vibrioid prosthecate dimorphic bacteria (strains ErkDOM-C, ErkDOM-E, and ErkDOM-YI). Frozen stocks of these three strains were streaked onto PYE and incubated for 6 days (23 °C) before inoculating single colonies into 25 ml PYE broth, followed by incubation at 23 °C, 200 rpm. Cells were harvested after three days from 8 ml exponentially growing culture ($OD_{600} ≈ 0.2$; cultures had been back-diluted with fresh PYE, after two days, to maintain exponential growth), and genomic DNA was extracted using Genomic tips 20/G (Qiagen) as described by Hallgren et al.[79] Long-read genome sequencing (Sequel system, Pacific Biosciences) and genome assembly was done at the National Genomics Infrastructure in Uppsala (Sweden). For this, DNA was sheared using Megaruptor 3 (Diagenode), and fragments below 10 kb were removed

using a PippinHT (Sage Science). The sequencing library was prepared using the SMRTbell Express Template Prep Kit 2.0 (Pacific Biosciences, USA) and sequenced on the Sequel single-molecule real-time cell platform (8 M v3, 4 cells; Pacific Biosciences, USA). Sequencing of strains ErkDOM-C, ErkDOM-E, and ErkDOM-YI yielded 99,762 reads totaling 2,973,558,276 bp, 120,349 reads totaling 3,547,634,714 bp, and 99,713 reads totaling 2,975,639,866 bp, respectively. The reads were assembled using Flye[80] v2.8.3 (default parameters). For strain ErkDOM-C, this resulted in a 4,064,099 bp genome (31× mean coverage depth), comprising a circular chromosome contig (3,918,693 bp) and a circular plasmid contig (145,406 bp). For strain ErkDOM-E, this resulted in a genome comprising a single circular chromosome contig (4,052,756 bp; 42× mean coverage depth). For strain ErkDOM-YI, this resulted in a 4,064,094 bp genome (32× mean coverage depth),

comprising a circular chromosome contig (3,918,688 bp) and a circular plasmid contig (145,406 bp). The genomes were made publicly available on GenBank (BioProject accession: PRJNA1228543; assemblies: GCA_048541825.1, GCA_048541895.1, and GCA_048541805.1).

## Selection of genomes

An initial set of 768 *Caulobacterales* genomes was collected as follows (Supplementary Data 1a). First, *Caulobacterales* MAGs were retrieved from metagenomes sequenced from Lake Erken (Sweden) microbial model community cultures[78] (*n* = 8). Additionally, genomes from *Caulobacter* strains ErkDOM-C, ErkDOM-E, and ErkDOM-YI (*n* = 3) isolated from one of the Lake Erken microbial model community cultures were included (see above). Lastly, genome assemblies identified as *Caulobacterales* in the Genome Taxonomy Database (GTDB) Release 07-RS207[81] (genomes released by July 12th, 2021) were retrieved accordingly from NCBI[82] RefSeq and GenBank databases in April 2022, with the exception of 20 genomes that were suppressed (*n* = 747). Of these genomes, 65 corresponded to MAGs annotated as *Caulobacterales* in the stratfreshDB[64] of stratified freshwater metagenomes. Thus, 58% (65/112) of the stratfreshDB *Caulobacterales* MAGs passed GTDB quality checks. Additional assemblies annotated as *Caulobacteraceae* and released after the GTDB release were also retrieved from NCBI (*n* = 10).

The completeness and contamination of these genomes were then estimated using the CheckM[83] v1.1.3 "lineage_wf" method and the "taxonomy_wf" method with the *Caulobacterales* order-specific dataset (except for alphaproteobacterial outgroup genomes, where the alphaproteobacterial class-specific dataset was used), which relies on Prodigal[84] for protein prediction and HMMER[85] for identifying marker genes. The 573 genomes with an estimated completeness ≥ 90% and contamination ≤ 5% ("taxonomy_wf") were selected for further analysis (Supplementary Data 1a). The genomes were then de-replicated to obtain species clusters at 95% average nucleotide identity (ANI) using dRep[86] v3.2.2 with the option "-sa 0.95". This resulted in selected genome representatives of 207 *Caulobacteraceae* species clusters and 140 outgroup *Caulobacterales* species clusters (including GCF_019455445.1, which was misclassified as *Caulobacteraceae* in NCBI) (Supplementary Data 1a). Changes were made to the automatically selected genome representatives whereby they were replaced with the species type strain, if one was available and had not been selected as the representative (applies to *B. aurantiaca* DSM 4731[T], *B. bullata*, HAMBI 262[T], *B. denitrificans* TAR-002[T], *B. diminuta* ATCC 11568[T], *B. huaxiensis* 090558[T], *B. vesicularis* NBRC 12165 [T], *C. vibrioides* DSM 9893[T] [CB51[T]], *He. mobilis* M65[T], *He. pelagia* LA220[T], *Hy. atlantica* 22II1-22F38[T], "*Hy. pacifica*" T16B2[T], *Hy. polymorpha* PS728[T], and *M. maris* DSM 4734[T]). An additional five *Alphaproteobacteria* genomes were included as an outgroup. This set of *Caulobacterales* genomes is here referred to as the "core dataset"

## Freshwater metagenome analyses, assigning *Acaudatibacter* gen. nov. species clusters, and defining the "extended dataset" of genomes

Data for the presence and absence in freshwater metagenomes, as well as relative abundance estimations, of *Caulobacteraceae* species clusters and *Acaudatibacter* gen. nov. species clusters, were subset from the dataset of Rodríguez-Gijón et al.[37] (Supplementary Data 1b). In short, the dataset comprises 80,561 prokaryotic genomes of medium-to-high-quality (>50% completeness and <5% contamination) according to CheckM[83] v1.1.3 "lineage_wf". The estimated genome size of all 80,561 genomes was calculated by dividing the assembly size by the estimated completeness provided by CheckM, and the genomic guanosine and cytosine (G + C) content was provided by CheckM. All medium-to-high quality genomes were grouped into species clusters at ANI ≥ 95%, using FastANI[87] v1.34 and mOTUpan[88] v0.3.2, obtaining 24,050 species clusters in total, each being represented by the highest-quality genome of the species cluster (i.e., the species representative).

This dataset was then mapped against a dataset of 636 short-read metagenomes from globally distributed freshwater environments using Strobealign[89] v0.11.0, detecting the presence of 9028 species in at least one metagenome. Further details can be found in Rodríguez-Gijón et al.[37] The subset of *Acaudatibacter* gen. nov. genomes available in this dataset is here referred to as the "extended dataset" (Supplementary Fig. S7 and Supplementary Data 1b), and was annotated and analyzed further. With regards to which *Acaudatibacter* gen. nov. genomes are represented, the "extended dataset" principally differs from the "core dataset" of genomes in three regards: (i) it also includes genomes from Nayfach et al.[90], (ii) it also includes newly re-binned stratfreshDB[64] MAGs (see Rodríguez-Gijón et al.[37]), and (iii) it had a less stringent genome quality threshold (>50% completeness and <5% contamination, instead of the >95% completeness and <5% contamination threshold used for the "core dataset").

## Genome annotation

Protein sequences (retrieved April 2022) and genome annotations (GenBank flat files; retrieved January 2024) were downloaded from NCBI[82] where available. In cases where they were unavailable for newly generated MAGs and genomes, Prokka v.1.14.6 (with the options "--compliant", "--kingdom Bacteria", "--gcode 11", "--addgenes") was used to call proteins (using Prodigal[91]). Resulting protein sequences were assigned to KEGG orthologs[92] (KOs) using GhostKOALA[93] v2.2 (v3.0 was used for annotations presented in Supplementary Fig. S16) and to eggNOG[94] v5.0 non-supervised orthologous groups (NOGs), as well as KOs, using eggNOG-mapper (emapper)[95] v2.1.5 (v2.1.12 was used for annotations presented in Supplementary Figs. S8, S16, and S25) (with the options "-m diamond" and "--sensmode more-sensitive"). Overviews of NOG and KO presence across genomes were generated using custom scripts (https://github.com/jennahd/anno-utils), and can be found in Supplementary Data 10–12. Pathway completeness was estimated using KEGG Decoder[96] v1.3 from GhostKOALA KO predictions (Supplementary Data 13). Protein domains were predicted using InterProScan[97] v5.52.-86.0 (with the options "-appl Pfam, TIGRFAM, PANTHER", "-iprlookup", and "-pa"). Proteins were further annotated with top hits from NCBI's nr database[82] (April 2022) using DIAMOND[98] blastp v2.0.9 (with the options "--max-target-seqs 1", and "--more-sensitive"). All protein annotations were then compiled using a custom script (https://github.com/jennahd/anno-utils). Reciprocal best blast hits[99,100] (RBH) against the *C. crescentus* CB15 assembly GCF_000006905.1 were determined using blastp[101] v2.15.0+ "-max_target_seqs 1 -evalue 1e-3" for protein sequences from each genome using custom scripts (https://github.com/j-hallgren/make-RBH) and can be found in Supplementary Data 14. To map locus IDs for the two *C. crescentus* laboratory strains CB15 and NA1000 ("CC" and "CCNA" numbers, respectively), RBH was also run for CB15 against the NA1000 genome assembly GCF_000022005.1. Pairwise ANIs were calculated using FastANI[87] v1.33 with default options (Supplementary Data 15). Pairwise average amino acid identities (AAI) were calculated using EzAAI[102] v1.2.3 'extract' and 'calculate' modules with default options (Supplementary Data 16). Genome statistics were calculated using Quast[103] v5.0.2, dRep[86] v3.2.2 (as outlined above), and CheckM[83] v1.1.3 (with the methods "lineage_wf" and "taxonomy_wf", as outlined above). Gene synteny maps were plotted using the R package gggenomes[104] v1.0.0 and edited in Adobe Illustrator 2025 for clarity. For this, gene coordinates were retrieved from GenBank flat files (.gbff) from NCBI, or from the GenBank files (.gbk) provided by Prokka when necessary, and syntenic regions were identified using blastp[101] v2.15.0+ (with the options "-evalue 1", "-task blastp-fast", and "-max_target_seqs 1") and through manual curation. For *B. subvibrioides* ATCC 15264[T], the *pufB* gene was manually drawn based on the coordinates from GCF_000144605.1 assembly version 07/24/2021. progressiveMauve[105] (Mauve snapshot_2015-02-25 build 0) was used to align *Acaudatibacter* PGC contigs.

## Species phylogenies

An initial phylogenetic tree was inferred using GToTree[106] v1.6.20 with the included set of 117 alphaproteobacterial single-copy marker genes. In short, the pipeline predicted genes with Prodigal[84] v2.6.3, then target genes were identified with HMMER3[85] v3.3.2, aligned individually with Muscle[107] v5.1.linux64, trimmed with TrimAl[108] v1.4.rev15, and then concatenated into a supermatrix alignment. An ML phylogeny was then inferred using IQ-TREE[109] v1.6.12 with the LG + C60 + F + G model of evolution, selected using ModelFinder[110] from LG profile mixture models (C10 to C60)[111], with or without empirical base frequencies (+F), and with 1000 ultrafast bootstraps[112] (Supplementary Fig. S2).

A manually refined ML species phylogeny was then inferred using the AlphaCOG dataset of 72 single-copy marker genes compiled by Martijn et al.[18] Protein sequences corresponding to each of the 72 COGs were extracted based on eggNOG NOG annotations. Individual ML gene phylogenies were inferred for each of the 72 marker genes by aligning with MAFFT[113] L-INS-i v7.407 (poorly aligned sequences were manually removed), trimming with trimAl[108] v1.4.1, and tree inference using IQ-TREE[109] v2.2.0 with models selected using ModelFinder[110] as outlined above for the initial species phylogeny, but including Gamma distributed (+G) or FreeRates (+R) models. Resulting phylogenies were visualized using a custom script (https://github.com/jennahd/tree-utils), and putative paralogs, contamination, long-branching, horizontal transfers, and duplicate sequences were manually removed (Supplementary Data 17). Individually, gene phylogenies were then inferred again as above and manually inspected. Trimmed alignments were then concatenated into a supermatrix alignment using PhyKIT[114] v1.11.7 with the "create_concat" option. A concatenated ML species phylogeny was then inferred as outlined for the individual gene phylogenies. An additional phylogeny was inferred using the posterior mean site frequency[115] (PMSF) approximation of the selected model (LG + C60 + F + R) with 100 non-parametric bootstraps (Fig. 1a and Supplementary Fig. S1).

An additional species phylogeny was inferred with the "extended dataset" of putative *Acaudatibacter* gen. nov. (GTDB taxon "g_Palsa-881") genomes to confirm their phylogenetic placement and to infer their species relationships (Supplementary Figs. S7a, b, Supplementary Data 1b). The following genomes were included: representatives of the twelve *Acaudatibacter* species clusters from the "extended dataset" (Supplementary Figs. S7a, b and Supplementary Data 1b), all 54 *Caulobacteraceae* species type strains of the "core dataset" (Supplementary Data 1a), the 16 species representatives from uncharacterized *Caulobacteraceae* genus-level clades of the "core dataset", and three outgroup *Caulobacterales* species type strains for rooting. A set of 117 alphaproteobacterial single-copy marker genes was retrieved, and a concatenated alignment was generated using GToTree[106] v1.8.6, as outlined above. An ML species phylogeny was then inferred using IQ-TREE[109] v1.6.12 with the LG + C60 + F + G model of evolution and 1000 ultrafast bootstraps[112] (Supplementary Fig. S9).

A species phylogeny was also inferred based on concatenated 16S and 23S rRNA genes to provide further insights into the phylogenetic relationships between *Acaudatibacter* gen. nov., *Caulobacter*, and *Phenylobacterium*. rRNA genes were extracted from both the species genome representatives of the "core" and "extended" datasets (Supplementary Data 18) using Barrnap v0.9 (https://github.com/tseemann/barrnap) with a cutoff of 50% of each gene ("--reject 0.5"), and using bacterial rRNA gene models ("--kingdom bac"). Based on long branches in initial phylogenies, the five alphaproteobacterial outgroup genomes and three genomes (GCA_014763715.1, *Rhodobacterales* bacterium DE15.006; GCA_016791445.1, *Hyphomonadaceae* bacterium new MAG-150; GCF_014323565.1, *Hyphobacterium* sp. CCMP332) with divergent rRNA genes were removed from the analysis. The remaining 16S (n = 220) and 23S (n = 203) rRNA gene sequences were individually aligned using MAFFT E-INS-i[113] v7.407 and trimmed with trimAl[108] v1.4.1 using the "-gappyout" option. 16S and 23S rRNA genes from the

same genome were then concatenated using PhyKIT[114] v1.11.7. An ML phylogeny was then inferred using IQ-TREE[109] v2.2.2.6 with the GTR + I + R7 model of evolution, selected using ModelFinder[110] from GTR models[116] and with 1000 ultrafast bootstraps[112] (Supplementary Fig. S3).

## Gene alignments and phylogenies

Protein sequence alignments of EAL (Pfam domain PF00563) and GGDEF (Pfam domain PF00990) domain-containing proteins (Supplementary Fig. S10) were done using Clustal Omega[117] and visualized in Jalview[118] ver.2.11.4.1 using the "Clustal" color option. For gene phylogenies, protein sequences were retrieved from the *Caulobacterales* species representative dataset corresponding to CreS (KEGG KO K18642 and/or Pfam domain PF19220), BchY (KEGG KO K11334 and/or PF00148 (BchYZBN)), and PufM (KEGG KO K08928 (PufL), K08929 (PufM), and/or Pfam domain PF00124). Initial gene phylogenies were then inferred together with corresponding Pfam domain sequences from the Pfam[47] database to identify which sequences from the species representatives corresponded to which gene homologs. Sequences were aligned using MAFFT[113] L-INS-i v7.407, poorly aligned positions trimmed using trimAl[108] v1.4.1 with the "-gt 0.2" option, and ML phylogenies inferred with IQ-TREE[109] v2.2.0 using the LG + C10 + G + F model of evolution. Sequences corresponding to each gene family of interest (CreS, BchY, or PufM) were extracted and then searched against NCBI's nr database[82] using DIAMOND[98] blastp v2.0.9 with the options "--more-sensitive" and "--max-target-seqs 2000". Unique hits with greater than 25% sequence identity and 30% coverage were collected using EDirect[119] v15.1 "efetch". Sequence redundancy of hits was reduced to 80% using CD-HIT[120] v4.8.1. Sequences were combined with the Pfam and species representative sequences and aligned using FAMSA[121] v2.2.2, poorly aligned positions were trimmed using trimAl[108] 1.4.1 with the "-gt 0.2" option, sequences with less than 30% coverage of the resulting alignment were removed using a custom script (https://github.com/jennahd/tree-utils), and ML phylogenies were inferred with IQ-TREE[109] v2.2.0 using the LG + C10 + G + F model of evolution and with 1000 ultrafast bootstraps[112]. Sequences from nr falling in the clades of interest, together with sequences from *Caulobacterales* species representatives, were selected (the original non-truncated sequences corresponding to the Pfam domain reference sequences were also captured by the nr search and they were removed from further analyses). Sequences with a "partial" keyword in the sequence header were removed, and alignment was performed using MAFFT[113] L-INS-i v7.407. Sequences that were poorly aligned, or that appeared truncated, were manually removed, and alignments were rerun. Trimming was then performed using trimAl[108] 1.4.1 with both the "-gt 0.5" and "-gappyout" options. ML phylogenies were then inferred using IQ-TREE[109] v2.2.0 with model selection using ModelFinder[110] from LG, WAG, JTT, and Q.pfam profile mixture models (C20, C40, and C60), and up to 10 FreeRate categories, and with 1000 ultrafast bootstraps[112]. For CreS, an additional phylogeny was inferred as outlined above using only sequences from the clade with members encoding the Crescentin Pfam protein domain (PF19220). The Q.pfam+C60 + R9, JTT + C60 + R5, WAG + C60 + R10, and WAG + C60 + R9 models of evolution were selected for CreS_ALL (all sequences), CreS_SUBSET (the subset described above), BchY, and PufM, respectively.

## Metadata collection and environmental distribution

Sampling metadata were manually collected between May and June 2024 from NCBI BioSample (https://www.ncbi.nlm.nih.gov/biosample/) and BioProject (https://www.ncbi.nlm.nih.gov/bioproject/) databases, as well as from listed publications and the JGI GOLD platform (https://gold.jgi.doe.gov/) when necessary, and can be found in Supplementary Data 2. Genomes were manually classified into simple broad environmental categories, based on the available sampling information. A meta-analysis of previous 16S rRNA gene amplicon studies was done using the Integrated Microbial Next-Generation Sequencing Platform (IMNGS)

database[25], with a *Taxonomy* job submitted on April 8, 2022, using the query "Bacteria/Proteobacteria/Alphaproteobacteria/Caulobacterales/Caulobacteraceae", for which the results can be found in Supplementary Data 3.

## Statistics and data visualization

Phylogenetic trees, including protein domains (formatted from InterProScan output using a custom script: https://github.com/jennahd/tree-utils), were visualized with iTOL[122] and annotated in Adobe Illustrator 2025. Plots were generated using the R[123] v4.4.0 packages ggplot2, gggenomes, and maps in Rstudio[124] v2024.04.0.735, GraphPad Prism v10.3.0, and Microsoft Excel for Mac v16.92, and edited in Adobe Illustrator 2025 for clarity.

## Reporting summary

Further information on research design is available in the Nature Portfolio Reporting Summary linked to this article.

## Data availability

Genome data used in this work is publicly available and was obtained from NCBI RefSeq (https://www.ncbi.nlm.nih.gov/refseq/), NCBI Genbank (https://www.ncbi.nlm.nih.gov/genbank/), the JGI Genome Portal (https://genome.jgi.doe.gov/portal/), Lake Erken model community MAGs (https://doi.org/10.17044/scilifelab.19923161.v1), and from sequenced Lake Erken *Caulobacter* isolates made available at Genbank (BioProject accession: PRJNA1228543; assemblies: GCA_048541825.1, GCA_048541895.1, and GCA_048541805.1). Additional data supporting the findings of this work is provided in the Supplementary Information, and as a tabular data file containing Supplementary Data 1–4, 6–7, and 10–18, as a PDF file containing Supplementary Data 5, 8 and 9, and as three video files containing Supplementary Movies 1–3. Additional raw data files, including genome annotations, sequence alignments, and phylogenetic tree files, are publicly available at Researchdata.se hosted by the Swedish National Data Service (https://doi.org/10.58141/0bz5-dc62). Source data are provided with this paper.

## Code availability

Custom scripts used for genome annotation and phylogenetic analyses are available on GitHub (https://github.com/jennahd/anno-utils and https://github.com/jennahd/tree-utils). Additional custom scripts used to run RBH analysis are available on GitHub (https://github.com/j-hallgren/make-RBH).

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

## Acknowledgements

We thank Emilie Falconer for assistance with microscopy, as well as Christine Jacobs-Wagner and Martin Thanbichler for sharing strains and plasmids. Genomic analyses and phylogenetic inferences were enabled through resources provided by the Swedish National Infrastructure for Computing (SNIC) at UPPMAX through the storage project SNIC 2021/6-99 and the computation project SNIC 2022/5-137, partially funded by the Swedish Research Council through grant agreement no. 2018-05973. We acknowledge support of the National Genomics Infrastructure (NGI)/Uppsala Genome Center and UPPMAX for providing assistance in massive parallel sequencing and computational infrastructure. Work performed at NGI/Uppsala Genome Center has been funded by RFI/VR and Science for Life Laboratory, Sweden. The work was financially supported by major funding from the Strategic Research Areas (SFO) program/Science for Life Laboratory distributed through Stockholm University to K.J. and S.L.G., and project grants from Stockholm University–Region Stockholm (FoUI-992787) and the Swedish Research Council (2020-03545; 2024-04942) to K.J. The work was further supported by an international postdoc grant from the Swedish Research Council (VR grant 2022-06250) and a non-stipendary EMBO Long Term Fellowship (ALTF 740-2022) to J.E.D.

## Author contributions

J.H.: conceptualization, methodology, software, formal analysis, investigation (all experimental work, comparative genomics, taxonomic descriptions, and meta-analyses), data curation, writing—original draft, writing—review and editing, and visualization. J.E.D.: conceptualization, methodology, software, formal analysis, investigation (all phylogenetic analyses), data curation, writing—review and editing, and funding acquisition. A.R.-G.: formal analysis, investigation (freshwater metagenome analyses), and writing—review and editing. J.N.: conceptualization and investigation. S.G.: conceptualization, resources, writing—review and editing, supervision, project administration, and

funding acquisition. K.J.: conceptualization, resources, writing—original draft, writing—review and editing, supervision, project administration, and funding acquisition.

## Funding

## Competing interests
The authors declare no competing interests.
