## [Transparent Peer Review file · Nature Communications]

Widespread potential for phototrophy and convergent reduction of lifecycle complexity in the dimorphic order *Caulobacterales*

Corresponding Author: Professor Kristina Jonas

Version 0:

Reviewer comments:

Reviewer #1

(Remarks to the Author)

The manuscript by Hallgren et al. presents a thorough analysis of the diversity and ecology of Caulobacterales merging in-silico and experimental approaches.

This is a monumental effort providing a wide range of data that will be of great interest to the microbiology community and form a solid basis for years of experimental work ahead. Notable results are: the multiple reversions from a complex dimorphic lifestyle that might be seen as an evolutionary dead end into a classical monomorphic one; the wider than expected distribution of these bacteria in variable environments; the potential phage origin of crescentin; the first evidence of photosynthesis and carbon fixation potential.

The manuscript is dense but well written and the data clearly presented, including a rich section of supplementary figures and full data, in particular sequence and accession numbers.

I only have a few minor comments:

Figure 1. What is normally done to ease reading a complex phylogenetic tree is to replace bootstrap values "higher than a certain threshold" with circles. Here from the legend it seems to me that the authors have replaced "all" BS by circles. Please amend or clarify. In the corresponding Figure S1, please replace the circles with real BS values.

Line 90: please say where these genomes were retrieved from. I suppose NCBI? Please indicate the month and year, as these databases change rapidly.

Line 128: Please add where the metadata were extracted from.

Line 162: why "unexpectedly" if the strain is immotile? Please clarify.

Lines 166-167: unclear, please reword.

Lines 203-204: the paper is already rich in results, but I kind of expected a minimum of annotation of these putative candidate flagellar genes, for example say how many have no annotation whatsoever, if they are clustered, or any specific domain category.

General remark concerning gene absences, did the author confirm by resequencing or PCR on the strains they use? As these genes are usually clustered, the readers have to be reassured that this is not due to assembly problems (I don't think so but maybe?).

One final remark is that I was confused between Sup Figures Sup Data and Extended Data, can the authors use a single type or clarify?

(Remarks on code availability)

Reviewer #2

(Remarks to the Author)

Major Comments

The manuscript presents a wide-ranging study of the ecological and evolutionary relationships within the order Caulobacterales, and how these genetic relationships inform our understanding of cell physiology in the model organism *Caulobacter vibrioides* (and vice versa). The authors conduct a series of targeted bioinformatic analyses focused on key distinguishing features of this clade, including cell dimorphism, morphology, division, and envelope structure. They also tracked down new leads about the diversity of energy metabolisms in Caulobacterales, providing new evidence for phototrophy throughout the clade, as well as photoautotrophy in a newly proposed sibling clade (genus), which is named in honor of a pioneering bacteriologist and champion of *Caulobacter* research (Dr. Jeanne Poindexter).

I found the analyses to be rigorous, appropriately scoped, and very detailed, aiding in the reproducibility and impact of this work. It was not a superficial or shotgun-style bioinformatic study. The authors develop clear, testable expectations through comparative genomics and, where possible, validate their findings through bench experiments, including a thoughtful gene complementation assay and photopigment characterization. These validations reinforce the strength of their phylogenomic inferences and demonstrate the rigor of their approach, meeting the high standards for scientific evidence expected by Nature Communications.

The work also meets the bar for impact, with the analysis spanning multiple dimensions of microbiology, addressing a deep need for resolving eco-evo questions. The study touches upon systematics (naming and shuffling several groups), cell biology, and the environmental distributions linked to genomic signature of niche. Perhaps one of its most understated but important contributions is the identification of lineages in which canonical dimorphic development has been lost. These lineages offer a rare opportunity to dissect how regulatory and structural pathways are rewired following the loss of highly conserved traits (across Caulobacteraceae). As the authors note: "Together this suggests that these species still maintain c-di-GMP metabolism, but that they might use it for regulating other functions than their core developmental programs." This observation opens exciting new questions about the plasticity and repurposing of regulatory systems, which can draw upon a large body of evidence from *C. vibrioides*.

I have several recommendations and questions posed in line comments that the authors should address. One critical part is to provide additional lines of evidence about the phylogenetic placement of the *Acaudatibacter*, which is of central importance to some of their arguments. Generally speaking my feedback supports possible improvements to the scope and impact of findings. I could not find any glaring flaws.

Overall, the authors present their findings clearly and with high quality visualizations to make complex data highly accessible. This study significantly advances our understanding of Caulobacterales. I find the work rigorous, well-documented, and impactful and would like to endorse its publication in Nature Communications.

Line Comments

L28: "species relationships" <- I believe "phylogenetic relatedness" may be more unambiguous since a 'relationship' between species could refer to a form of symbioses.

L112: Excellent choice in proposed name. It is about time! (That said Dr. Poindexter hails from the Midwest where there are few mountains to speak of)

L198 – 205: Strong exploratory analysis and hypothesis generation.

L235: It might be worth noting here whether the S-layer genes differ between the terrestrial-associated and aquatic-associated *Caulobacter*. This appear to be the case, but reporting a measure of this would provide further insights, building on earlier descriptions.

L267: This may be a bit in the weeds, but I'm curious whether you observed any patterns in the distribution or genomic context of *lovK*, *lovR* in relation to genes associated with phototrophy. These proteins are encoded by *C. vibrioides* (e.g., PMC3370868) and can be photoactivated. Given you have all the bioinformatic capabilities to query at your fingertips, I'd be curious to know if you find anything related to this.

L366: "...despite lacking flagellar genes, perhaps by regulating buoyancy through pilus-mediated cellular aggregation, as previously reported for other anoxygenic phototrophs." <- The authors should consult the work by Dr. Feibig on cell physiology required for *C. vibrioides* to concentrate at the neuston (<https://journals.asm.org/doi/full/10.1128/jb.00064-19>). They write: "Unlike the holdfast, neither the flagellum nor type IV pili are required for *C. crescentus* to partition to the air-liquid interface." In lieu of this result, could the authors take a second look to see any if any patterns in holdfast-related genes stand out in the phototrophic lineages? Figure 7A shows that absence of holdfast adhesins, but I recommend taking a second, closer look.

L369: The only issue I have with the Discussion is that the authors might highlight that the loss of developmental machinery coincides with a reorganization of the c-di-GMP signalling for cell development. This restructuring of regulatory mechanisms might be just as interesting from a model organism perspective. Does it make sense to highlight the potential for *Acaudatibacter* or *P. immobile* to serve as a comparative model for *Caulobacter*?

L610: The phylogenetic placement of *Acaudatibacter* is central to several of the authors' claims. While the phylogenetic methods used appear robust and the branch support values are strong, I recommend supplementing this analysis with an

ANI-based tree. As a gene-agnostic approach, ANI (recommend FastANI; <https://www.nature.com/articles/s41467-018-07641-9>) could provide an independent line of evidence to confirm the placement of the Acaudatibacter lineage.

L783: Did you include hfi (holdfast inhibitor) genes in your set to query? If not, why?

(Remarks on code availability)

Version 1:

Reviewer comments:

Reviewer #2

(Remarks to the Author)

I am satisfied with the responses provided by the authors. They have thoughtfully addressed all criticism and have provided evidence in their manuscript to support their decision to focus on more promising and interesting genomic features based on their analyses. Their work holds up to scrutiny and I am confident it will have an impact.

(Remarks on code availability)

Point-to-point response to reviewers (manuscript # NCOMMS-25-34606-T)

We thank the reviewers for their helpful feedback and time spent assessing our work. We have carefully considered all comments and have addressed them by modifying the manuscript text to improve the clarity of our paper, and in one case by adding new data. Please find our responses below in blue text, with the original reviewers' comments in black text. Please also note that we have renumbered the supplementary material to make it compatible with the *Nature Communications* formatting requirements. The line numbers, supplementary material, and figure references provided by us in this point-to-point response refer to the revised manuscript.

Reviewer #1 (Remarks to the Author):

The manuscript by Hallgren et al. presents a thorough analysis of the diversity and ecology of Caulobacterales merging in-silico and experimental approaches.

This is a monumental effort providing a wide range of data that will be of great interest to the microbiology community and form a solid basis for years of experimental work ahead. Notable results are: the multiple reversions from a complex dimorphic lifestyle that might be seen as an evolutionary dead end into a classical monomorphic one; the wider than expected distribution of these bacteria in variable environments; the potential phage origin of crescentin; the first evidence of photosynthesis and carbon fixation potential.

The manuscript is dense but well written and the data clearly presented, including a rich section of supplementary figures and full data, in particular sequence and accession numbers.

Response 1. We thank the reviewer for this positive assessment.

I only have a few minor comments:

Figure 1. What is normally done to ease reading a complex phylogenetic tree is to replace bootstrap values "higher than a certain threshold" with circles. Here from the legend it seems to me that the authors have replaced "all" BS by circles. Please amend or clarify. In the corresponding Figure S1, please replace the circles with real BS values.

Response 2. The circles shown in **Fig. 1a** do indeed represent non-parametric bootstrap support (npBS) values, as mentioned in the figure legend and as illustrated in the top-left corner of the figure panel. However, all npBS values are not shown with circles. Black circles indicate $npBS = 100$, gray circles indicate $90 \leq npBS < 100$, and white circles indicate $70 \leq npBS < 90$. Thus, branches with npBS values below 70 lack circles altogether. We thank the reviewer for highlighting that this might be a source of

confusion. We have clarified in the figure legend that the circles are color-coded. In the corresponding **Supplementary Fig. S1**, all npBS values are already written out as numbers, as proposed by the reviewer.

Line 90: please say where these genomes were retrieved from. I suppose NCBI? Please indicate the month and year, as these databases change rapidly.

Response 3. We agree that the information regarding database names and date of retrieval are critical. Our genomes have been sourced from NCBI RefSeq, NCBI GenBank, the dataset of Garcia *et al.* (2022, <https://doi.org/10.1101/2022.03.21.485019>), and from three newly isolated *Caulobacter* strains (this work). How genomes were collected, from which databases and datasets the genomes were retrieved, as well as the date of data retrieval, are described under the **Methods** subheading “**Selection of species genome representatives**”. Since the specific details regarding genome sourcing are important, but rather lengthy and therefore potentially distracting to the reader if presented in the main text, we have now added “(see **Methods**)” to the end of the sentence in question to help guide the reader to this key information. To further help with clarity, we have modified the **Methods** subheading “**Selection of species genome representatives**” into “**Selection of genomes**”.

Line 128: Please add where the metadata were extracted from.

Response 4. This information is available in the **Methods** section, under the subheading “**Environmental distribution**”. Similar to **Response 3**, since the specific information on the metadata source is rather long, instead of specifying the information in the main text, we again added “(see **Methods**)” to the end of this sentence to help direct the reader to the relevant information. To further help with clarity, we have modified the **Methods** subheading “**Environmental distribution**” into “**Metadata collection and environmental distribution**”.

Line 162: why "unexpectedly" if the strain is immotile? Please clarify.

Response 5. Immotility phenotypes observed under laboratory conditions can be the result from low flagellar gene expression caused by the choice of growth medium or other selected laboratory conditions, and do not necessarily mean that the strain lacks the genetic potential for motility altogether. Moreover, immotility phenotypes can be the result of small mutations, such as the pseudogenization, truncation, or deletion of a single key flagellar gene. However, what we find for *Phenylobacterium immobile* and the *Acaudatibacter* spp. must be the result of a large number of mutations driven by a considerable selection pressure. These lineages lack orthologs of ~50 motility genes (flagellar/chemotaxis components and regulators), in comparison to their closest relatives. Moreover, since several of these orthologs are typically represented in the genomes by numerous paralogs, the total number of genes absent is even higher. See for example the high paralog copy numbers of the chemotaxis gene *mcp* in

Supplementary Data S5 (upwards of > 20 copies). The flagellar/chemotaxis genes are normally spread out over the chromosome (see **Response 8**), meaning that their absence is likely the result of a large number of mutations driven by a considerable selection pressure. Importantly, this absence of genes stands out against other known immotile isolates, such as the type strains of *P. hankyongense* and *P. soli*, which have large numbers of flagellar genes despite being immotile, as emphasized at the end of the paragraph to which the reviewer referred. We therefore found it striking and surprising to find the complete absence of flagellar genes from *P. immobile* and the *Acaudatibacter* spp.

However, we agree with the reviewer that “unexpectedly” has the risk of being perceived as a paradoxical statement, given that *P. immobile* have been described as immotile. We therefore have changed “unexpectedly” to “strikingly”, which still highlights the significance of the finding.

Lines 166-167: unclear, please reword.

Response 6. We agree with the reviewer that our previous phrasing was unclear and thank them for bringing this to our attention. We have modified the sentence to clarify that the *Acaudatibacter* genomes of the “extended dataset” were subset from the dataset of Rodríguez-Gijón (2025; <https://doi.org/10.1101/2025.03.24.644981>).

How this “extended dataset” differs from our “core dataset” of genomes has now also been clarified further in the **Methods** section, under the section which we have renamed ‘**Freshwater metagenome analyses, assigning *Acaudatibacter* gen. nov. species clusters, and defining the “extended dataset” of genomes**’. Principally, the “extended dataset” contains more *Acaudatibacter* species clusters than our “core dataset” due to:

- It also includes genomes from the dataset of Nayfach *et al.* (2021; <https://doi.org/10.1038/s41587-020-0718-6>)
- They re-binned the stratfreshDB (Buck *et al.* [2021; <https://doi.org/10.1038/s41597-021-00910-1>]) database of metagenomes
- They used a less stringent genome quality threshold. (> 50% completeness and < 5% contamination, instead of the > 95% completeness and < 5% contamination thresholds that we used for our “core dataset”).

Lines 203-204: the paper is already rich in results, but I kind of expected a minimum of annotation of these putative candidate flagellar genes, for example say how many have no annotation whatsoever, if they are clustered, or any specific domain category.

Response 7. We did indeed exclude such details from the manuscript due to the scope and large amount of data of the paper. However, to better highlight some of the diversity regarding their predicted [and lack of predicted] functions, as requested by the reviewer, we have modified the sentence as follows (new text underlined): “These

genes encode putative signaling proteins, transcriptional regulators, enzymes, and hypothetical proteins, and are promising candidate genes for potential flagellar motility and/or cellular dimorphism factors.”

General remark concerning gene absences, did the author confirm by resequencing or PCR on the strains they use? As these genes are usually clustered, the readers have to be reassured that this is not due to assembly problems (I don't think so but maybe?).

Response 8. Designing meaningful PCR tests to validate gene losses, like those observed in *P. immobile*, is much more technically challenging than a simple PCR validation of the presence of a gene. This is because it is difficult to design meaningful primer pairs without prior knowledge of which loci were affected by the gene losses and how the loci were affected. And such analyses are further precluded by the fact that *P. immobile* is a rather long-branching lineage, currently lacking genome-sequenced close relatives, which makes it more challenging to analyze exactly how flagellar loci were mutated during their loss.

However, there are several lines of evidence that reassure us that the observed gene/presence absence patterns are very unlikely to be mere assembly artifacts:

1. The many flagellar, chemotaxis, holdfast, and developmental genes absent from these lineages are normally spread out over the chromosome (as mentioned in the **Discussion**). As an example of how evenly spread out these genes typically are over the bacterial chromosome, in the *C. crescentus* CB15 genome, we find at least 15 chemotaxis/flagellar gene clusters and 19 loci with orphan chemotaxis/flagellar genes (**Revision Table 1**), not to mention the numerous loci that encode cell developmental regulators and holdfast proteins (**Supplementary Data S14a**). As is evident from the locus numbers listed in **Revision Table 1**, the flagellar/chemotaxis loci are spread out evenly over the circular chromosome (the locus IDs span CC_0001 to CC_3763). The fact that the genomes of *P. immobile* and the *Acaudatibacter* spp. are completely devoid of specifically flagellar/chemotaxis and developmental genes is therefore very improbable to be an assembly artifact. It is here worth mentioning that the *P. immobile* genome assembly is composed of only 6 contigs.
2. The 32 *Acaudatibacter* metagenome-assembled genomes (MAGs) that show the specific absence of flagellar/chemotaxis/development/holdfast genes are derived from diverse environments and have been assembled independently in various sequencing projects (**Supplementary Fig. S7**).
3. For *Acaudatibacter* species clusters that are represented by multiple genome assemblies, the gene presence/absence patterns are consistent across the assemblies (**Supplementary Fig. S8**). For example, *Ac. aquilonius* is represented by a total of twelve MAGs from four different freshwater bodies from two countries (**Supplementary Fig. S7**), which all show very similar gene presence/absence patterns (**Supplementary Fig. S8**).

4. The *Acaudatibacter* species that specifically lack flagellar motility, chemotaxis, holdfast and development genes form a monophyletic clade (**Supplementary Fig. S9**).
5. The gene presence/absence patterns are consistent with phenotypic data for *P. immobile*, such as its lack of motility as well as flagella and other cellular appendages (e.g., holdfasts and flagella) as observed by Lingens *et al.* (1985; <https://doi.org/10.1099/00207713-35-1-26>) and its symmetric mode of reproduction observed in our work (**Fig. 3**).

Thus, taken together, it is highly unlikely that the remarkably similar gene presence/absence pattern seen between *P. immobile* and *Acaudatibacter* spp. is a spurious pattern that has resulted from an assembly artifact. Instead, it is more likely that the observed gene patterns are the result of convergent evolution. For all these reasons we do not believe that it is necessary to re-sequence *P. immobile* E^T or to design PCR tests to test for gene absences.

To help reassure the reader, we have modified the first paragraph of the **Results** section “**Convergent losses of dimorphism-related genes in *Caulobacteraceae***”, most notably by adding the following sentence: “Given the large number of independently sampled and assembled *Acaudatibacter* genomes specifically lacking these genes, and that motility genes are normally spread out over the chromosome, the observed gene absence patterns are unlikely to stem from assembly artifacts.”

One final remark is that I was confused between Sup Figures Sup Data and Extended Data, can the authors use a single type or clarify?

Response 9. The manuscript formatting largely followed the style of another journal, to which we originally submitted the manuscript. We have now updated the formatting to match the guidelines of Nature Communications, by for example removing the distinction between “Extended Data Figures” and “Supplementary Figures”.

Revision Table 1. List of flagellar and chemotaxis gene loci in *C. crescentus* CB15, showcasing how widespread they normally are over the bacterial chromosome. *mcp*: homologs of methyl-accepting chemotaxis protein.

Type	Locus IDs	Notable genes	#
Flagellar/chemotaxis gene clusters	CC_0428–0441	cheYI, cheAI, cheWI, cheRI, cheBI, cheYII, cheD, cheV, cleA, cheE	1
	CC_0589–0598	mcp, cheY, mcpK, cheAII, cheWII, cheYII, cheBII, cheRII	2
	CC_0792–0794	fljMNO	3
	CC_0898–0910	fljgL, flaN, flgDE, sciP, fliFG, flbE, fliN, flbD, flhA	4
	CC_0951–0955	fliPO, flgBC, fliE	5
	CC_1075–1077	fliQR, flhB	6
	CC_1456–1465	flmH, flbAT, flaF, fljLK, fljJ, flaY	7
	CC_1907–1908	flhB, fliK	8
	CC_2058–2066	fssB, motE, pfII, fliML, flgFGAH	9
	CC_2581–2584	fliX, flgl, flbY	10
	CC_2839–2847	flgE, mcp, mcp	11
	CC_2854–2857	flmD, flmEF	12
	CC_3040–3041	fliJ	13
	CC_3258–3259	cheYIII, cheE	14
	CC_3471–3472	cheYIV, cheRIII	15
Putative orphan flagellar/chemotaxis genes	CC_0066	mcp	16
	CC_0161	mcp	17
	CC_0343	mcp	18
	CC_0504	mcp	19
	CC_0750	motA	20
	CC_0764	cheW	21
	CC_1399	mcp	22
	CC_1573	motB	23
	CC_1655	mcp	24
	CC_2175	fliN	25
	CC_2281	mcp	26
	CC_2317	mcp	27
	CC_2691	mcp	28
	CC_2810	mcp	29
	CC_2976	flgL	30
	CC_3025	cheW	31
	CC_3145	mcp	32
	CC_3349	mcp	33
	CC_3358	mcp	34

Reviewer #2 (Remarks to the Author):

Major Comments

The manuscript presents a wide-ranging study of the ecological and evolutionary relationships within the order Caulobacterales, and how these genetic relationships inform our understanding of cell physiology in the model organism *Caulobacter vibrioides* (and vice versa). The authors conduct a series of targeted bioinformatic analyses focused on key distinguishing features of this clade, including cell dimorphism, morphology, division, and envelope structure. They also tracked down new leads about the diversity of energy metabolisms in Caulobacterales, providing new evidence for phototrophy throughout the clade, as well as photoautotrophy in a newly proposed sibling clade (genus), which is named in honor of a pioneering bacteriologist and champion of *Caulobacter* research (Dr. Jeanne Poindexter).

Response 10. We would like to clarify that *Poindextera* gen. nov. does so far not include members with phototrophic potential. *Acaudatibacter* gen. nov is the genus containing species with photoautotrophic potential.

I found the analyses to be rigorous, appropriately scoped, and very detailed, aiding in the reproducibility and impact of this work. It was not a superficial or shotgun-style bioinformatic study. The authors develop clear, testable expectations through comparative genomics and, where possible, validate their findings through bench experiments, including a thoughtful gene complementation assay and photopigment characterization. These validations reinforce the strength of their phylogenomic inferences and demonstrate the rigor of their approach, meeting the high standards for scientific evidence expected by Nature Communications.

The work also meets the bar for impact, with the analysis spanning multiple dimensions of microbiology, addressing a deep need for resolving eco-evo questions. The study touches upon systematics (naming and shuffling several groups), cell biology, and the environmental distributions linked to genomic signature of niche. Perhaps one of its most understated but important contributions is the identification of lineages in which canonical dimorphic development has been lost. These lineages offer a rare opportunity to dissect how regulatory and structural pathways are rewired following the loss of highly conserved traits (across Caulobacteraceae). As the authors note: “Together this suggests that these species still maintain c-di-GMP metabolism, but that they might use it for regulating other functions than their core developmental programs.” This observation opens exciting new questions about the plasticity and repurposing of regulatory systems, which can draw upon a large body of evidence from *C. vibrioides*.

I have several recommendations and questions posed in line comments that the authors should address. One critical part is to provide additional lines of evidence about the phylogenetic placement of the *Acaudatibacter*, which is of central

importance to some of their arguments. Generally speaking my feedback supports possible improvements to the scope and impact of findings. I could not find any glaring flaws.

Overall, the authors present their findings clearly and with high quality visualizations to make complex data highly accessible. This study significantly advances our understanding of Caulobacterales. I find the work rigorous, well-documented, and impactful and would like to endorse its publication in Nature Communications.

Response 11. We are very grateful for the kind feedback provided by the reviewer.

Line Comments

L28: “species relationships” <- I believe “phylogenetic relatedness” may be more unambiguous since a ‘relationship’ between species could refer to a form of symbioses.

Response 12. We agree with the reviewer and have made the proposed change to the Abstract.

L112: Excellent choice in proposed name. It is about time! (That said Dr. Poindexter hails from the Midwest where there are few mountains to speak of)

Response 13. We are glad to hear that the reviewer finds the name fitting.

L198 – 205: Strong exploratory analysis and hypothesis generation.

Response 14. We thank the reviewer for this positive feedback.

L235: It might be worth noting here whether the S-layer genes differ between the terrestrial-associated and aquatic-associated *Caulobacter*. This appear to be the case, but reporting a measure of this would provide further insights, building on earlier descriptions.

Response 15. We could not find any correlation between the presence and absence of genetic potential for the RsaA S-layer among *Caulobacter* species and the environmental type, i.e., between aquatic freshwater versus plant-associated terrestrial species (**Revision Figure 1**). As noted in the manuscript in the **Results**, in the first paragraph of the “**Most Caulobacterales members lack genetic potential for S-layers and crescentin-mediated cell curvature**” section, and as shown in **Fig. 4b**, the genetic potential for the RsaA S-layer is essentially restricted to the major *Caulobacter* clade that includes *C. segnis* and *C. vibrioides/crescentus*. This clade comprises both freshwater and terrestrial lineages. Similarly, and notably, the presence and absence of the cell curvature-inducing cytoskeletal protein crescentin does also not correlate with the environmental type (**Revision Figure 1**). We chose not report this lack of correlation in the manuscript. However, we would like to point

out that comparisons like these can already be made by the reader by cross-referencing the data available in the figures of the manuscript. Moreover, other similar types of analyses can readily be made by the reader by utilizing our datasets (e.g., **Supplementary Data S2** and **S11–S14**).

To briefly allude to this lack of correlation between the presence/absence of S-layer genes and the environmental type, we have modified the sentence describing the presence of S-layer genes accordingly (new text underlined): “Specifically, the S-layer genes *rsaADEF* were limited to the environmentally widespread (see **Fig. 1a,b**) *C. crescentus*–*segnis* clade [...]”.

Revision Figure 1. Comparison of the phylogeny and environmental distribution to the presence and absence of genes for the RsaA S-layer and the curvature-inducing cytoskeletal protein crescentin across *Caulobacter* species. Presented as in **Supplementary Fig. 6a**. Gene annotations were retrieved from **Supplementary Data S11a**.

L267: This may be a bit in the weeds, but I'm curious whether you observed any patterns in the distribution or genomic context of *lovK*, *lovR* in relation to genes associated with phototrophy. These proteins are encoded by *C. vibrioides* (e.g., PMC3370868) and can be photoactivated. Given you have all the bioinformatic capabilities to query at your fingertips, I'd be curious to know if you find anything related to this.

Response 16. We agree that the possibility for photosynthesis genes being regulated by the light-sensing LovKR module is an interesting prospect. Looking at the reciprocal best blast hit (RBH) dataset (**Supplementary Data S14a**) as a proxy for the presence/absence of LovR and LovK, we found no clear patterns when comparing genomes with or without the genetic potential for phototrophy (**Revision Figure 2a**). This lack of correlation was also the case when running blastp (v2.14.1+) searches for LovR (CC_0284) and LovK (CC_0285) across our *Caulobacterales* dataset (**Revision Figure 2b**). Moreover, we have manually searched through the genome annotations (available in the Figshare data repository at: Core-dataset_Gene_Annotations/all_genomes.annotations.tar.gz) for the genomes with phototrophic potential of our dataset, but never found LovK or LovR homologs in proximity to any photosynthesis gene clusters (PGCs) or orphan phototrophy genes. Any potential links between LovKR and the PGC therefore remain obscure. However, it is worth noting that our results have revealed that the PGCs of *Caulobacterales* species consistently harbor *ppsR* (which like *lovK* encodes a PAS-domain S-box protein) and *ppaA/aerR* homologs (**Supplementary Fig. S15**), master regulators of the PGC that among other things are involved in light sensing (Vermeulen and Bauer, 2016; *J. Bact.*; <https://doi.org/10.1128/JB.00374-15>). In *Rhodobacter* spp. PpsR binds DNA at specific binding sites and acts as a repressor under high light and/or aerobic conditions. PpsR is in turn controlled by the antirepressor PpaA/AerR, which binds a tetrapyrrole and acts as a light sensor. This system is thus in some ways analogous to LovKR.

Revision Figure 2. Lack of correlation of the presence/absence of LovKR and holdfast genes with phototrophic potential. Phylogeny and phototrophy markers correspond to Fig. 1. (a) Reciprocal best blast hit (RBH) analysis of the “core dataset” of *Caulobacterales* genomes using LovR (CC_0284) or LovK (CC_0285) from *C. crescentus* CB15 as query. Blue = presence, light-gray = absence. Gene annotations were retrieved from **Supplementary Data S14a**. (b) Log10-transformed E-values from blastp v2.14.1+ analysis (default parameters) of the “core dataset” of *Caulobacterales* genomes, using LovR (CC_0284) or LovK (CC_0285) from *C. crescentus* CB15 as query. Light-gray represents absence of a blast hit. “INF” represents E-values = 0, which could not be log10-transformed. (c) Holdfast gene annotations retrieved from **Supplementary Data S5**. Blue = presence [RBH], dark-gray = presence [KEGG KO], light-gray = absence.

L366: “...despite lacking flagellar genes, perhaps by regulating buoyancy through pilus-mediated cellular aggregation, as previously reported for other anoxygenic phototrophs.” <- The authors should consult the work by Dr. Feibig on cell physiology required for *C. vibrioides* to concentrate at the neuston (<https://journals.asm.org/doi/full/10.1128/jb.00064-19>). They write: “Unlike the holdfast, neither the flagellum nor type IV pili are required for *C. crescentus* to partition to the air-liquid interface.” In lieu of this result, could the authors take a second look to see any if any patterns in holdfast-related genes stand out in the phototrophic lineages? Figure 7A shows that absence of holdfast adhesins, but I recommend taking a second, closer look.

Response 17. In contrast to *C. vibrioides* which is an obligate aerobe that forms pellicles at the air–liquid interface, our results show that the photoautotrophic *Acaudatibacter* spp. exhibit specific localization patterns at high relative abundance at the upper-most parts of the anoxic (lower) freshwater stratum (the hypolimnion) (**Fig. 6c**). We believe that there might be fundamental differences in the dynamics that are at play when a microorganism positions itself at the air–liquid interface compared to the liquid–liquid interface at the oxycline (the epilimnion–hypolimnion interface). For the former, the buoyancy of the microorganism principally only needs to be high enough that it remains floating. However, for the latter, the level of buoyancy needs to be fine-tuned, perhaps dynamically, since too high buoyancy would bring the microorganism into the low-density oxic epilimnion and too low buoyancy would cause sedimentation. For this reason, we are unsure how extrapolatable the findings in *C. vibrioides* (Fiebig, 2019) are to the *Acaudatibacter* spp. with photoautotrophic potential, regarding the essentiality of the holdfast for air–liquid interface localization.

Regarding the patterns in holdfast-related genes, we cannot find any particular patterns for the presence/absence of holdfast-related genes. More specifically, there are both phototrophic lineages with (e.g., *Brevundimonas subvibrioides*, *B. bacteroides*, *Oceanicaulis satelles* comb. nov. [*Alkalicaulis satelles*], *Aquidulcibacter paucihalophilus*) and without (e.g., phototrophic *Hyphomonas* spp., *Acaudatibacter* spp., and “CAIMFV01” spp.) genes for holdfast adhesins (**Revision Figure 2**). It is worth noting that several species with phototrophy genes are known holdfast-producers, which validates their genetic potential for holdfast production, e.g., *B.*

bacteroides, *B. subvibrioides*, and *B. variabilis* (Poindexter, 1964; *Bacteriol. Rev.*; <https://doi.org/10.1128/br.28.3.231-295.1964>). Holdfast-mediated cellular rosettes are for example visible for *B. variabilis* and *B. bacteroides* in **Supplementary Fig. S4**. Thus, as we point out in the final paragraph of the **Discussion**, despite their shared metabolic potential, phototrophic *Caulobacterales* species are developmentally and morphologically diverse, something that is perhaps reflected in the variation of their genetic potential for holdfast adhesins.

L369: The only issue I have with the Discussion is that the authors might highlight that the loss of developmental machinery coincides with a reorganization of the c-di-GMP signalling for cell development. This restructuring of regulatory mechanisms might be just as interesting from a model organism perspective. Does it make sense to highlight the potential for *Acaudatibacter* or *P. immobile* to serve as a comparative model for *Caulobacter*?

Response 18. We agree that our findings regarding the prevalence of c-di-GMP-related modules among pathways being lost in *P. immobile* and *Acaudatibacter* species are important, but were not properly represented in the **Discussion**. We also agree with the reviewer that *P. immobile* (and indeed *Acaudatibacter* species, if future culturing attempts will be successful) represents a promising new model system for comparative cell biological studies. We have added a paragraph to the **Discussion** to better highlight these outcomes of our work.

L610: The phylogenetic placement of *Acaudatibacter* is central to several of the authors' claims. While the phylogenetic methods used appear robust and the branch support values are strong, I recommend supplementing this analysis with an ANI-based tree. As a gene-agnostic approach, ANI (recommend FastANI; <https://www.nature.com/articles/s41467-018-07641-9>) could provide an independent line of evidence to confirm the placement of the *Acaudatibacter* lineage.

Response 19. While it would be ideal to confidently resolve the phylogenetic placement of *Acaudatibacter* gen. nov., we do not think that its position relative to other genera, such as *Phenylobacterium* or *Caulobacter*, is essential to our key claims. The *Acaudatibacter* clade itself is always supported as a monophyletic group separate from *Caulobacter* and *Phenylobacterium*, and the apparent loss of genes involved in dimorphic traits occurred after the divergence of the genera. Regardless of the position of *Acaudatibacter* relative to *Caulobacter* and *Phenylobacterium*, there are two distinct lineages that show lack of dimorphism-associated genes, i.e., *P. immobile* and ten (out of twelve) *Acaudatibacter* species, both of which are always nested within clades that include deeper branching lineages that have the genes. The putatively non-dimorphic *Acaudatibacter* species consistently form a supported monophyletic 'crown-group' clade within *Acaudatibacter*, while the species *Ac. sp. SZAS_AMP-5* and *Ac. sp. CP_BM_RX_R9_33* branch basal to this crown group (**Supplementary Figs. S1, S2, and S9**) and encode large numbers of dimorphism-associated genes, exhibiting similar genetic potential for cell development as *C. crescentus* (see **Fig. 2b**), and thus strongly

suggest extensive gene loss in only the ‘crown-group’ *Acaudatibacter* lineage. In addition, the robust clustering of *P. immobile* within *Phenylobacterium* species encoding extensive suites of dimorphism-associated genes, reveals two independent incidences of large losses of dimorphism genes.

Regarding ANI, we had already run a full comparison of the pairwise ANI for all our genomes using FastANI (**Supplementary Data S15**). However, while ANI is a powerful metric for comparisons of closely related strains on the species and subspecies-level, it has limited resolution for deeper evolutionary relationships and is not suitable for comparisons on the genus level (Qin et al., 2014, <https://doi.org/10.1128/jb.01688-14>). Accordingly, phylogenetic signal is quickly lost even for within-genus comparisons in our ANI dataset; looking for example at *Phenylobacterium*, many pairwise ANI values within *Phenylobacterium* are below 80% and some are even far enough below 80% that FastANI simply outputs “NA” (**Revision Figure 3**). Thus, our ANI comparisons offered little insights into the deeper evolutionary relationships between the putative non-dimorphic lineages and their closest dimorphic relatives (**Revision Figure 3**). If anything, our ANI comparisons show that *P. immobile* has higher ANI to other species of *Phenylobacterium* than to *Acaudatibacter* species, supporting our major conclusions. But, given the limited utility of ANI for genus-level comparisons, we would not like to make a point out of this.

Revision Figure 3. Excerpt from **Supplementary Data S15**, showing the pairwise average nucleotide identities (ANIs) for *Phenylobacterium*, *Acaudatibacter* gen. nov., and *Caulobacter* genomes of the “core dataset”. Numbers show percentages. Black box borders demarcate genera. The yellow box border highlights *P. immobile*. White cells contain “NA” output values from FastANI, the result of an ANI value “much below 80%”, in other words ANI values so low that FastANI deems them uninformative (see FastANI documentation; <https://github.com/ParBLISS/FastANI>).

However, to supplement our protein sequence-based marker-gene species phylogenies, we have instead inferred a species phylogeny based on concatenations of the 16S and 23S rRNA genes. We have added this new phylogeny to the revised manuscript as the new **Supplementary Fig. 3** and updated the **Methods** section accordingly. First, it is worth noting that rRNA genes are not always easily recovered in metagenome-assembled genomes (MAGs) (Yuan et al., 2015, <https://doi.org/10.1093/bioinformatics/btv231>; Mise and Kawasaki, 2022, <https://doi.org/10.1038/s43705-022-00204-6>), thus out of the twelve *Acaudatibacter* species clusters (all of which are represented by MAGs), we were only able to detect rRNA genes (at > 50% completeness, using the tool Barnap) in four representative

Acaudatibacter genomes (**Revision Table 2**). Yet, despite the limited sampling of *Acaudatibacter*, the genus clustered with full support (100% ultrafast bootstrap support [ufBS]) sister to *Phenylobacterium* in the rRNA gene phylogeny (**Supplementary Fig. S3**), consistent with our more refined concatenated protein marker species phylogeny (**Supplementary Fig. S1**). Moreover, *Caulobacter* and *Phenylobacterium* were supported as monophyletic groups (100% ufBS). Importantly, mirroring our previous species phylogenies, *P. immobile* is nested within *Phenylobacterium*, clustering with species that have extensive suites of dimorphism-associated genes. Likewise, the putatively non-dimorphic *Acaudatibacter* species *Ac. boreus* Ki1-2-2m_bin-386^{Ts}, *Ac. sp.* 3300015024_21, and *Ac. sp.* 3300025574_7 cluster together with the basal species *Ac. sp.* CP_BM_RX_R9_33 which has a *C. crescentus*-like gene suite for dimorphic cell development (see **Supplementary Fig. S8**). Thus, this rRNA gene-based phylogeny further supports our conclusions regarding the convergent evolution of lifecycle complexity loss and adds additional support for the topology of *Acaudatibacter*, *Phenylobacterium*, and *Caulobacter* found in our refined marker protein species phylogeny.

Revision Table 2. 16S and 23S rRNA genes recovered using Barrnap from *Acaudatibacter* species representative genomes.

Species	Dataset	16S	23S
Ac. boreus Ki1-2-2m_bin-386 ^{Ts}	“core”	Partial (975 bp)	N/A
Ac. sp. CP_BM_RX_R9_33	“core”	Partial (909 bp)	N/A
Ac. sp. 3300015024_21	“extended”	Full length (1478 bp)	Full length (2798 bp)
Ac. sp. 3300025574_7	“extended”	Partial (947 bp)	Full length (2793 bp)

It is also worth mentioning here that in the new rRNA gene phylogeny we found that *Poindextera montana* comb. nov. S6^T clustered together with the *Acaudatibacter* representatives, although relationships within the clade were not supported (**Supplementary Fig. S3**). The clustering of *Poindextera* with *Acaudatibacter* was not observed in our marker protein-based phylogenies (**Supplementary Figs. S1, S2, and S9**). However, this clustering is perhaps not so unexpected given the limitations of the rRNA gene analysis: (i) we did not recover rRNA genes from members of the genus-level clade “CAISGS01” that *Po. montana* S6^T usually clusters with, and therefore it is not included in the rRNA gene tree, (ii) we only have a restricted representation of *Acaudatibacter* (**Revision Table 2**), and (iii) the rRNA gene phylogeny has much less information than the concatenated protein marker gene trees (alignment length of 4266 nucleotides, compared to the alignment lengths of 25448, 21766, and 23561 amino acids, respectively, for the trees in **Supplementary Figs. S1, S2, and S9**). Nonetheless, we find good support for the position of *Poindextera*, outside of the *Caulobacter–Phenylobacterium–Acaudatibacter* clade in our most refined protein marker gene species phylogeny (**Supplementary Fig. S1**), a position also recovered

in our other protein marker gene trees (**Supplementary Figs. S2 and S9**), which is therefore probably the more likely scenario given available data. Regardless, the alternative position of *Poindextera* does not affect the main conclusions of our work.

Moreover, the new rRNA gene phylogeny also provides further support for the clustering of *Aquidulcibacteraceae* fam. nov. as a coherent clade, separate from *Caulobacteraceae*, and sister to *Hyphomonadaceae* (**Supplementary Fig. S3**). We have thus added two sentences to the **Supplementary Discussion 2** section “**Motivation for the description of *Aquidulcibacteraceae* fam. nov.**”, to highlight this additional evidence. We have also added a brief reference to the new rRNA gene phylogeny (**Supplementary Fig. S3**) in the first paragraph of the **Results** section to highlight how the rRNA gene tree provides further support of our refined concatenated marker protein phylogeny (i.e., **Fig 1a** and **Supplementary Fig. S1**).

Lastly, to better emphasize that the lineages lacking dimorphic genes are always nested (as discussed in our first paragraph here in **Response 19**) within clades that have the genes, we have modified the first paragraph of the **Results** section “**Convergent losses of dimorphism-related genes in *Caulobacteraceae***” accordingly (new text underlined):

“Thus, *P. immobile* and most members (10/12) of the new genus *Acaudatibacter* lack flagellar genes and are both robustly nested within clades where deeper branching lineages have flagellar genes. Together, this strongly suggests that flagellar motility has been lost in at least two separate *Caulobacteraceae* lineages.”

L783: Did you include hfi (holdfast inhibitor) genes in your set to query? If not, why?

Response 20. We did indeed include the HfiA holdfast inhibitor as a query for our reciprocal best blast hit analysis. The presence and absence of this regulator is shown in **Fig. 2b** and **Supplementary Data S5**, as the right-most column in the “Cell cycle and developmental regulators” set of genes. Consistent with Berne (2023; PLOS Genet.; <https://doi.org/10.1371/journal.pgen.1010648>), we find that it has an extremely limited distribution across *Caulobacterales*, essentially only being detected in three *Caulobacter* species in total: *C. vibrioides* (both the type strain CB51^T and the lab model strain CB15), *C. sp.* CB13b1a, and *C. sp.* 602-1. We are not aware of the presence of additional *hfi* genes.